# A Study of Regional Assertions in the Architecture of Delhi from the 1970s to the present

**Sanyam Bahga \***  **and Gaurav Raheja**

Department of Architecture and Planning, Indian Institute of Technology Roorkee, Uttarakhand 247667, India; grdesfap@iitr.ac.in

**\*** Correspondence: sbahga@ar.iitr.ac.in; Tel.: +91-9876-870-121

**Abstract:** Critical regionalism is an architectural approach that seeks to correct sterile and abstract modernism by using contextual forces that focus on local needs and potential. As globalisation disrupts and displaces local building traditions in India's metropolitan cities, critical regionalism offers resistance to the homogenising forces of global modernism. This paper analyses five key architectural works realised in Delhi in the past four decades that incorporate the ideas of critical regionalism in their designs. The different approaches adopted by regionalist architects in dealing with local climate, topography, materials and sociological complexes have been presented. By limiting itself to regionalist works in Delhi, the paper attempts to highlight that critical regionalism is not a set of aesthetic preferences but a philosophical framework capable of producing diverse forms of architecture despite analogous external influences arising from similar site conditions.

**Keywords:** critical regionalism; modern regionalism; Indian architecture; Indian modernism; Delhi; local urban context; climate responsive architecture; socially-engaged architecture

## 1. Introduction

Contemporary architecture in India's metropolitan cities is being subjected to the homogenising forces of global capitalism [1]. The resulting 'corporate architecture' with its free-standing boxes wrapped in glass and aluminium façades makes little sense in India's climate and social conditions [2]. However, as the impact of globalisation has surged, there has been a greater inclination amongst local architects to resist its standardising forces by embracing the ideas of critical regionalism [3,4]. Practitioners of critical regionalism absorb the progressive lessons of modernisation and reconcile them with the natural forces and cultural realities of the place [5]. Thus, critical regionalism holds the promise to culturally enrich Indian architecture without hindering its advancement towards a bright and progressive future [6].

In this study, five critical regionalist projects realised in Delhi in the past four decades were identified and subsequently researched; their designs were analysed to assess how well they integrate the determinants of critical regionalism which the author has delineated as a result of this study. The different approaches adopted by different architects in response to the different aspects of critical regionalism have been presented. The philosophy of critical regionalism is rooted in local conditions such as climate, topography, architectural heritage, availability of materials and culture. By limiting itself to the Delhi region, the study attempts to highlight that despite similar site conditions resulting from close proximity of building sites, critical regionalism is able to generate diverse forms of architecture.

## 2. Critical Regionalism: Definition

The term 'critical regionalism' was first used by Uruguayan painter Pedro Figari in the 1920s [7]. The term first appeared in architectural discourse in essays by Alexander Tzonis and Liane Lefaivre

(1981) and Kenneth Frampton (1983) [8,9]. As an architectural approach, critical regionalism attempts to find the right balance between the universal lessons of science and locally rooted traditions of particular regions [10]. It is essentially a variant of modern architecture that shows greater respect towards the climate, topography, local materials and sociological complexes of a place [11] (p. 453). Critical regionalism is vital in contemporary times as it resists the homogenising force of global capitalism by using contextual forces that impart a sense of place and meaning to architecture.

*Critical Regionalism: Determinants*

Scholarship on critical regionalism reveals key determinants that an architectural work should assimilate in order to be critically regionalist:

1. Contextual response: The design should relate to its urban context by stressing upon certain site-specific factors [12].
2. Historical knowledge: The design should interpret lessons from traditional architecture of the region and blend them with contemporary architectural language [11] (p. 568).
3. Climate responsiveness: The form and organisation of the building should respond to the local climate minimising the need for mechanical controls like air-conditioning and artificial lighting [13] (p. 57).
4. Ecological sensitiveness: The design should minimise its impact on the natural environment of the place [13] (p. 29).
5. Local materials and construction: The design should make use of local materials and construction techniques that allow local workforce to participate in the construction process [14].
6. Technological sustainability: The design should adapt modern technology to meet the needs of the local community [15].
7. Cultural appropriateness: The architecture should take care of the way of life of the people who will be inhabiting the building [16].

Based on the determinants of critical regionalism, the study identified five architectural works realised in the Delhi metropolitan region in the past four decades that integrate the ideas of critical regionalism in their designs:

1. Yamuna Apartments (1980) by The Design Group;
2. Central Institute of Educational Technology (1988) by Raj Rewal;
3. India Habitat Centre (1993) by Joseph Allen Stein;
4. Development Alternatives Headquarters (2008) by Ashok B Lall;
5. Dilli Haat Janakpuri (2014) by Sourabh Gupta.

The above-mentioned architectural works have been analysed to assess how they integrate the determinants of critical regionalism in their designs. By analysing each determinant in each of the identified regionalist works, the study attempts to highlight that critical regionalism is not a formula or a fashion, but a process that authentically seeks to answer specific problems of a place. Moreover, by limiting itself to the Delhi region, the study tries to show that despite similar site conditions due to close proximity, critical regionalism is able to produce varied architectural responses due to the discretion it affords architects in selection of external influences and the final amalgamation between them.

## 3. Critical Regionalism in Delhi

### 3.1. Yamuna Apartments (1980)

The Yamuna Apartments are a low-rise medium-density housing scheme located in the Alaknanda neighbourhood of south-east Delhi (Figure 1). This cooperative group housing scheme was one of the first to be commissioned by the Delhi Development Authority (DDA). Although DDA had been one of the largest builders of housing in Delhi since 1961, it relied on byelaws and building codes that gave

little importance to architectural innovation [17]. Due to limitations arising from the building codes, the architects of Yamuna Apartments had to accommodate two hundred dwelling units in a small area of 3.75 acres with building height restricted to four-storey. Despite these limitations, the architects Ranjit Sabikhi and Ajoy Choudhury of the Design Group have been able to create an integrated community settlement by drawing upon principles from traditional built-forms. Sabikhi's education at the University of Liverpool and Choudhary's work experience in Milan brought a Western rigour to their design in form of clean lines and a minimalist appearance, while their years of upbringing and practice in the Delhi region has rooted their work in the tangible realities of the locale.

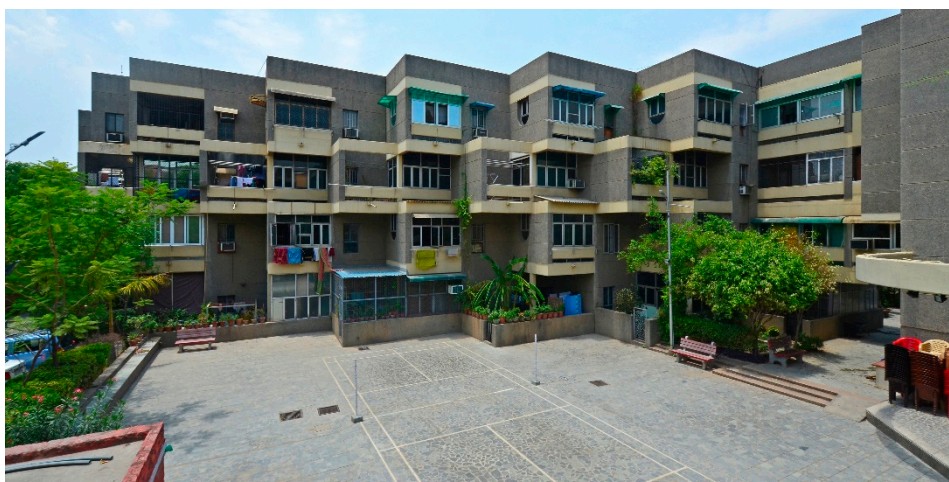

**Figure 1.** Yamuna Apartments, New Delhi (1980) by the Design Group (Photo: Author).

### 3.1.1. Contextual Response

The challenge faced by the architects in responding to the urban context of the site came from the fact that the Yamuna Apartments were the first housing scheme to be built in the Alaknanda neighbourhood of south-east Delhi. There was nothing existing around the site at the time to which the architects could respond to. Nonetheless, considering the allocation of neighbouring plots for future housing schemes (Figure 2), the design of Yamuna Apartments responds to the generalised modernist architectural vocabulary prevalent in urban India at the time. As a consequence, the Yamuna Apartments have been able to visually blend in with the housing developments in the neighbourhood that came up in the following years. However, the security concerns of the residents led to creation of a gated community that has compromised the linkages of the housing scheme with the surrounding neighbourhood [18].

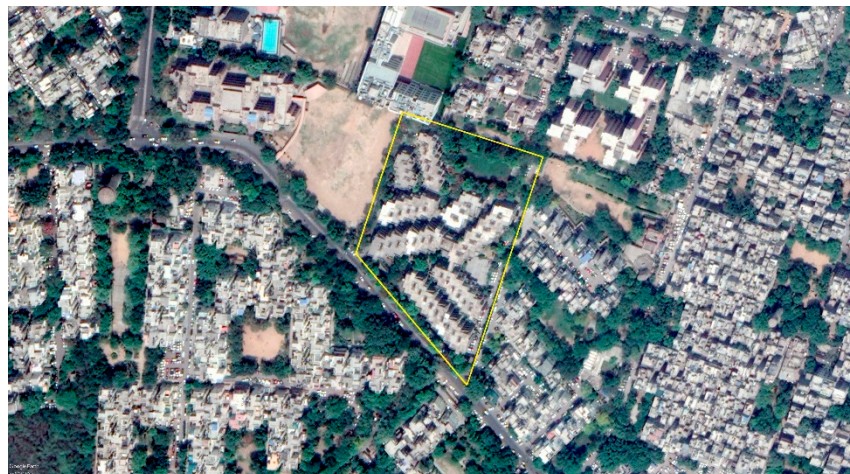

**Figure 2.** Satellite image of the Yamuna Apartments complex located in the Alaknanda neighbourhood of south-east Delhi (Image courtesy of Google Earth).

### 3.1.2. Historical Knowledge

The housing units of Yamuna Apartments have been grouped into eight building blocks. These blocks have been organised around a network of narrow pedestrian streets while the vehicular traffic is limited to the periphery of the site (Figure 3). The design concept draws upon the layout of traditional residential quarters of north Indian cities having a lively network of short streets where the residents can spend time sitting and mingling with their neighbours [19].

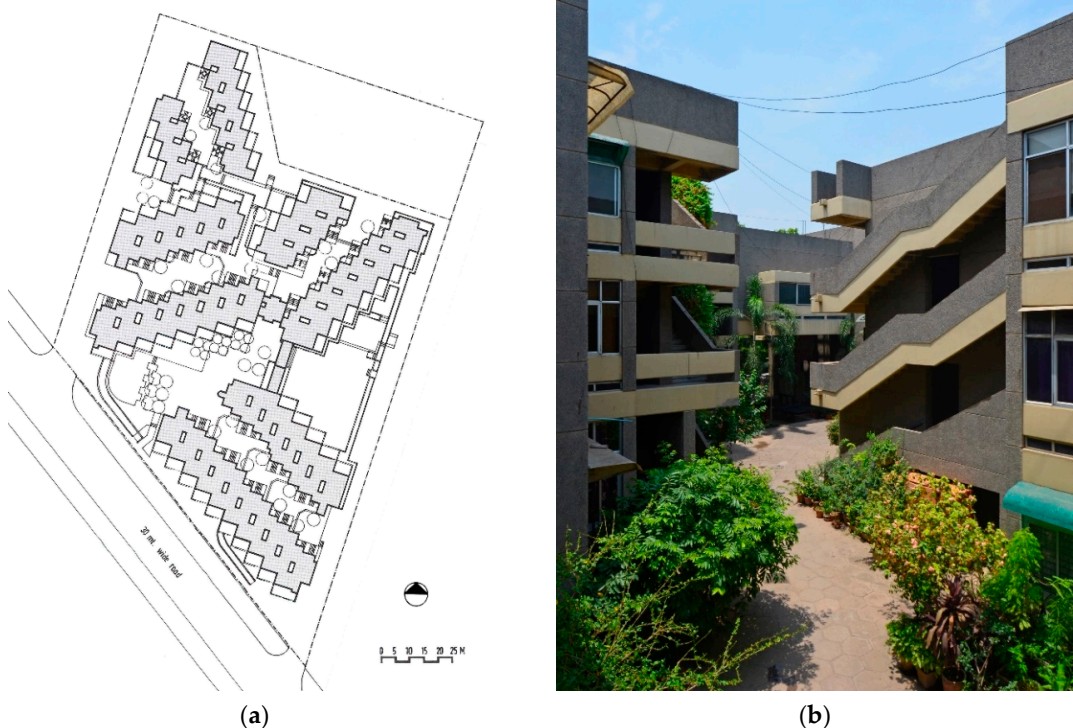

(**a**)    (**b**)

**Figure 3.** (**a**) Site Plan of Yamuna Apartments with housing blocks placed along four radially-converging streets [19]; (**b**) housing units overlooking the narrow pedestrian streets in Yamuna Apartments (Photo: Author).

In addition, these narrow pedestrian streets converge onto a central square in a manner similar to a traditional Indian village. In a typical village setting, the streets converge onto the central square that houses a market or a village well, thus acting as a place for communal interaction. A similar concept has been adopted in Yamuna Apartments as the central square containing recreational facilities, shops, a community club and a canteen forms the focus of the housing scheme.

Other elements assimilated from the region's traditional architecture include use of balconies for shading external walls from the harsh sun and provision of terraces for outdoor sleeping on hot summer nights.

### 3.1.3. Climate Responsiveness

As use of air-conditioning in households was quite uncommon in India in the 1970s, the architects of Yamuna Apartments prioritised cross-ventilation in unit designs to provide thermal comfort. Well-distributed openings on two sides of each unit have been supplemented by internal ventilation shafts to ensure adequate cross-ventilation (Figure 4). Space for installing desert coolers has also been provided in the internal ventilation shafts [20].

Other measures in response to the local climate include projection of deep balconies for protecting the external walls from incident sunrays. Square and rectangular-shaped balconies alternate between floors to break the monotony in façades and also to ensure sufficient daylighting in adjoining rooms.

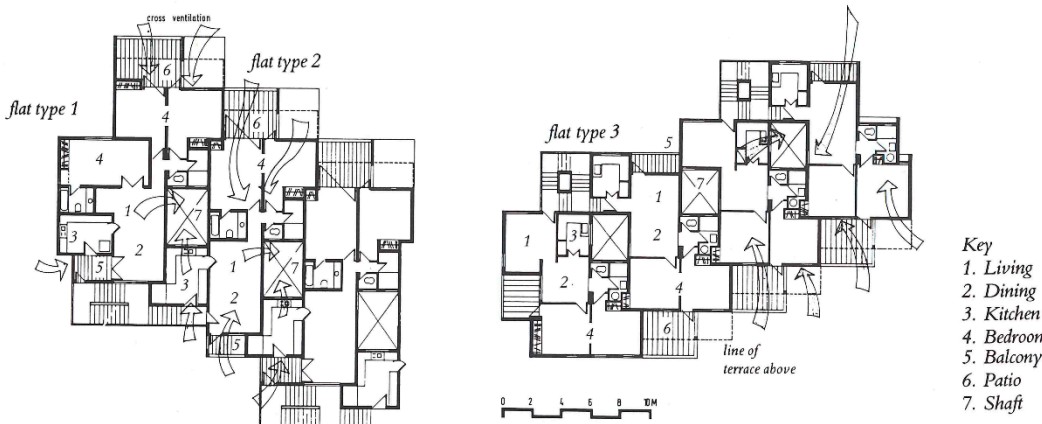

**Figure 4.** Floor plans of the three unit types with arrows depicting cross-ventilation [19].

### 3.1.4. Ecological Sensitiveness

The site located in south-east Delhi is flat and even, gently rising about one metre from the eastern edge to the western. Thus, the site did not pose a significant ecological concern for the architect. As mandated by the planning regulations applicable to the site, the architects had to leave an area of 0.5 acres in the north-eastern corner of the site as a green space (Figure 3). The remaining 3.75 acres of the site had to accommodate two hundred dwelling units with building height restricted to four-storey. Therefore, not much room was afforded by the building codes to preserve any existing vegetation on the site. Nonetheless, the architects have managed to provide sufficient vegetation cover along the internal streets in the site plan (Figure 5).

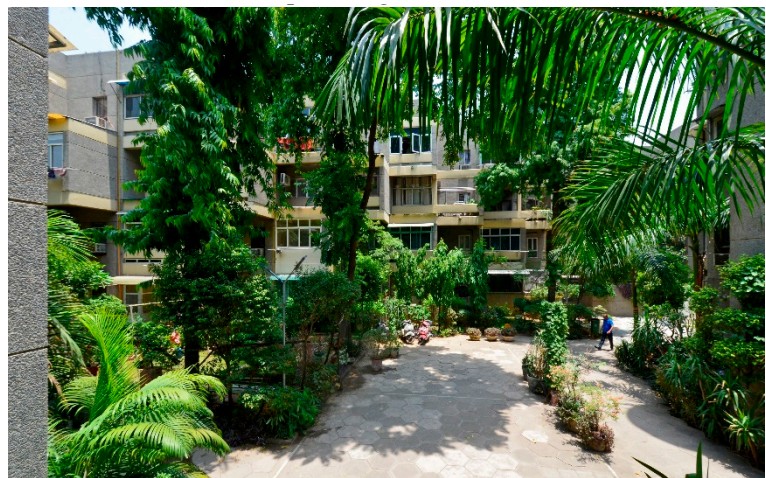

**Figure 5.** Vegetation cover along the pedestrian streets in Yamuna Apartments (Photo: Author).

### 3.1.5. Local Materials and Construction

The choice of materials for the Yamuna Apartments has been influenced by the exigencies of local economics. For the purpose of finishing the external walls, exposed stone aggregate plaster has been applied in situ. Aggregate plaster is an economical and maintenance-free material composed of simple ingredients: Cement and crushed local stone. Despite being an inexpensive material, stone aggregate plaster lends a rich granular texture to the building.

The structural components are fabricated in reinforced cement concrete, and cast in situ. Being labour-intensive, the in situ construction resulted in an increased involvement of local workforce in the construction process.

### 3.1.6. Technological Sustainability

Being a low-rise medium-density housing scheme, the Yamuna Apartments did not require considerable technological advancements for its construction. Nonetheless, wherever necessary, the architects have adopted the latest technologies of the time. Waffle slab has been employed to span the community club to furnish a column-free space. Besides, the housing scheme makes aggressive use of cantilevered balconies and staircases. This bold expression of structure, however, lends significant Brutalist overtones to the housing scheme, thus subduing its regional qualities.

### 3.1.7. Cultural Appropriateness

The design of Yamuna Apartments benefited from the fact that the architects knew beforehand the group of residents for whom they were designing for. The residents comprised a close-knit group of civil servants of South Indian origin who shared similar living requirements. Keeping this in mind, the architects have designed an integrated community settlement that takes its inspiration from traditional residential quarters of Indian cities. The housing blocks are compactly clustered around narrow pedestrian streets with the semi-private areas of housing units overlooking these streets. As the concept of privacy in Indian neighbourhoods is not as strong as it is in the West, each unit has been designed to open towards the pedestrian spine. This has resulted in creation of a convivial public space between the housing blocks for the residents to interact with and the children to play, undisturbed by the vehicular traffic.

The access staircase to each housing unit forms an extension of the street [21] (p. 153). The entrance to each house is through balconies that help in transitioning from the shared public space to the private areas of each unit. The internal layouts of units adhere to the traditional relationship between different rooms [22]. Each house is divided into a semi-private zone consisting of a living room, kitchen and dining room, and a private zone consisting of bedrooms and terraces (Figure 6). As all the residents share an orthodox Hindu lifestyle, due care has been taken in preserving the sanctity of the kitchen—considered a holy place—by keeping it away from the toilets. The semi-open terraces to the rear side are accessible from the bedrooms to facilitate the practice of outdoor sleeping during hot summer nights.

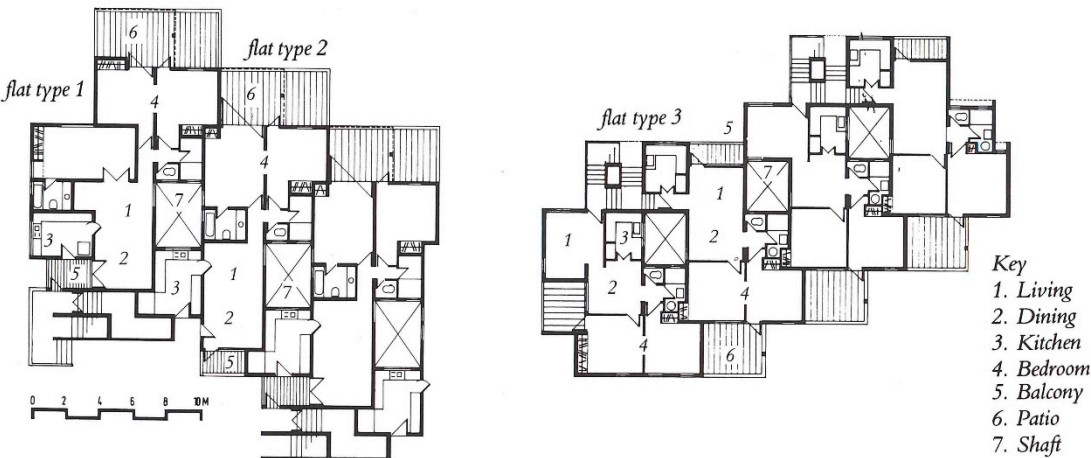

**Figure 6.** Floor plans showing the internal layouts of the three unit types [19].

### 3.2. Central Institute of Educational Technology (1988)

The Central Institute of Educational Technology (CIET) is a public-sector establishment primarily involved in multimedia production for use as educational tools (Figure 7). The building's 10,500 square metres of covered area comprises of television studios, sound studios, production rooms, seminar halls and projection areas along with ancillary facilities. Architect Raj Rewal makes use of traditional north Indian morphology to respond to this relatively unique institutional programme. His education at

the Architectural Association School in London and work experience in London and Paris instilled in him progressive lessons of modernisation, while his upbringing in Hoshiarpur, Punjab and Delhi has sensitized his work to the nuances of local culture.

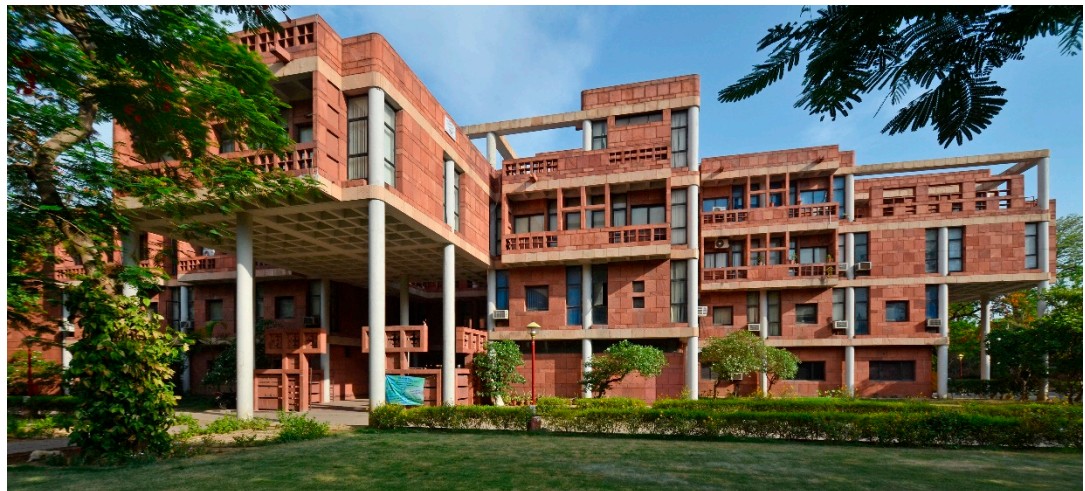

**Figure 7.** Central Institute of Educational Technology, New Delhi (1988) by Raj Rewal (Photo: Author).

### 3.2.1. Contextual Response

The CIET building is located on the southern edge of the 70-acre campus of the National Council of Educational Research and Training (NCERT, Figure 8). The various buildings situated within the campus have been designed by different architects and the only common feature between them is their height. Besides the CIET building, the other buildings in the NCERT campus do not have a strong architectural character as most of them are poor imitations of the exposed-concrete vertical slab buildings popularised in India by Le Corbusier through his Chandigarh Secretariat building. Therefore, instead of relating the design of the CIET building to buildings in the NCERT campus, Rewal chooses to respond to the national park and the historic Qutab Minar monument located on the south-west of the building. Framed views of the woodland and the distant Qutab Minar are provided from the upper level terraces [23]. Moreover, the use of red sandstone as primary cladding material is a common feature between the CIET building and the Qutab Minar.

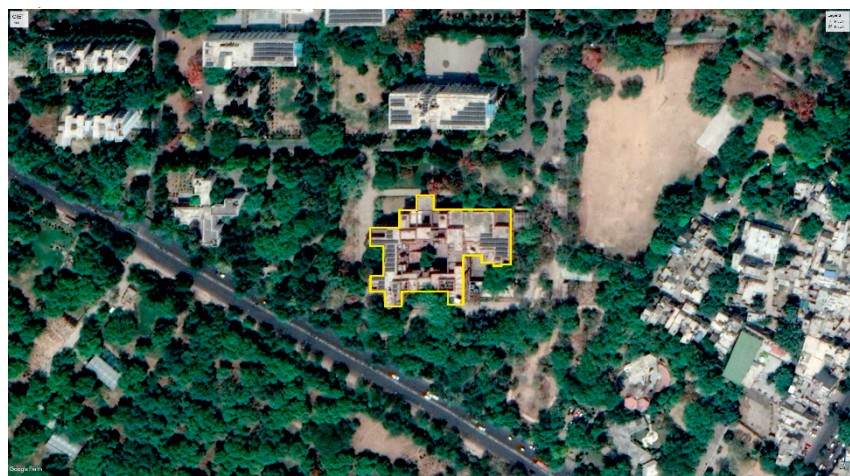

**Figure 8.** Satellite image of the Central Institute of Educational Technology (CIET) building located on the southern edge of the National Council of Educational Research and Training (NCERT) campus (Image courtesy of Google Earth).

### 3.2.2. Historical Knowledge

Rewal's architecture has been highly influenced by the urbanscape of historic Indian towns like Jaisalmer [22]. In the case of the CIET building, Rewal attempts to recreate the traditional street patterns of historic Indian towns within the circulation spaces of the building. Thus, the corridors on the upper levels frequently break into seating enclosures where the users can gather for socialising with each other (Figure 9).

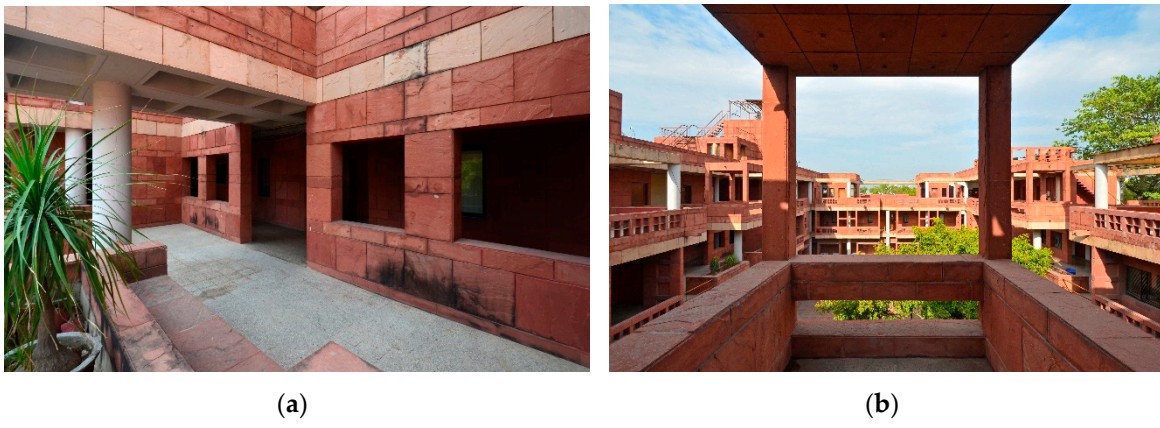

(**a**)                                                  (**b**)

**Figure 9.** (**a**) Seating enclosure adjoining the corridor on the second floor level of the CIET building; (**b**) view from underneath a *chhatri* (canopy) on the uppermost level of the CIET building (Photos: Author).

The central courtyard in the CIET building has corridors surrounding on all sides with enclosed rooms beyond (Figure 12). According to Rewal this setting has deliberately been influenced by the architecture of the madrasa (traditional Islamic school) [24].

Another device in the CIET building that seeks an indirect connection with the architectural heritage of the region is the contemporary *chhatri* (canopy). *Chhatris* have been provided on the terraces of the uppermost level to act as covered look-out points (Figure 9).

### 3.2.3. Climate Responsiveness

As Delhi lies in a climatic zone that is categorised as hot-and-dry for most part of the year, it is important to ensure that heat is delayed from reaching the interior of the building [25]. In the CIET building, Rewal has clad the exterior brick in-fill walls with sandstone thickening the external skin of the building and delaying the heat transfer into the building (Figure 10).

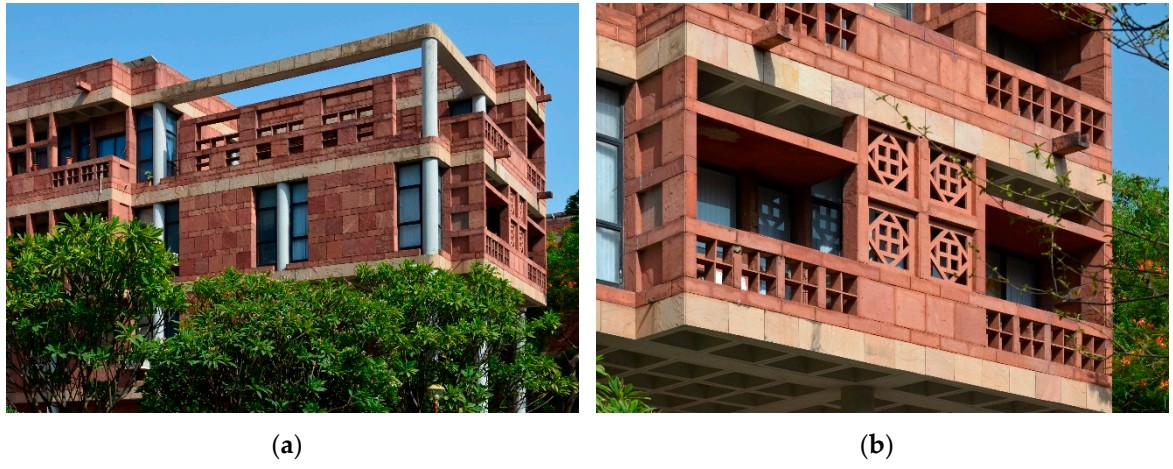

(**a**)                                                  (**b**)

**Figure 10.** (**a**) Sandstone clad external skin delays heat transfer into the CIET building; (**b**) balconies and *jaali* screens protect the rooms of the CIET building from direct sun (Photos: Author).

The central courtyard with a large shading tree within also helps in establishing a cooler micro-climate with improved humidity (Figure 11). The circulation corridors around the central courtyard are wide creating an additional buffer from the incident sunrays.

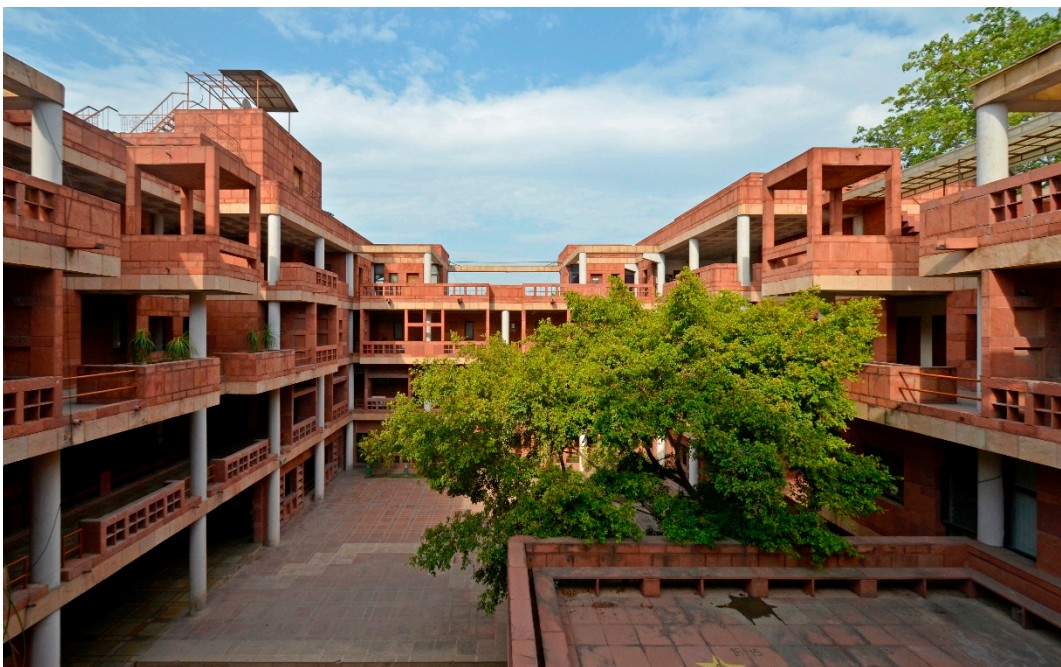

**Figure 11.** Central courtyard in the CIET building (Photo: Author).

On the outer edge of the building, deep balconies project out from the sides that are exposed to direct sunrays. Furthermore, *jaali* screens are provided on the balconies to protect the rooms from direct sun (Figure 10).

### 3.2.4. Ecological Sensitiveness

The site located within the NCERT campus is flat and even, and therefore did not create a major ecological concern for the architect. The challenge to the architect, however, arose from a large tree located in the middle of the site. Instead of treating the tree as a hindrance, Rewal turned it into a focal point of his design. He created a large courtyard around the tree and wrapped the built-up areas around this courtyard (Figure 11). The tree continued to flourish for years after the completion of the CIET building further proving that due care was taken to protect the tree and its roots during the construction process.

### 3.2.5. Local Materials and Construction

The predominant material used in the CIET building is Agra stone, a variety of red sandstone. The material is plentifully available in the region and has been historically used in prominent monuments in Delhi such as the Humayun's Tomb and the Red Fort. Other than red sandstone, Dholpur stone, a variety of yellow sandstone, is used for horizontal stripes on the building's exterior (Figure 10). For the purpose of flooring, marble mosaic and Kota stone, a greenish-blue variety of limestone, have been used. All these materials are abundantly available in the nearby Rajasthan state and have been used for construction purposes in Delhi for many centuries now.

The CIET building also benefits from local craftsmen for the production of *jaali* screens that enclose the balconies (Figure 10). Traditionally a *jaali* screen is a perforated stone latticed screen that provides shade from the sun while allowing air to flow through. Rewal, however, has modernised the intricate design of traditional *jaalis* into a more geometric pattern achieved by modern tools. Thus, local craftsmen not only got involved in the production phase, they also advanced their skills [26].

### 3.2.6. Technological Sustainability

Throughout his work, Rewal has relied heavily on the structural abilities of technological advancements in the building industry [27]. This approach has served him well in creating high-rise structures such as the State Trading Corporation building (1989) in New Delhi as well as large-span structures like the Pragati Maidan Permanent Exhibition Complex (1972), also in New Delhi.

The design of the four-storeyed CIET building, however, did not require considerable technological advancements. Nonetheless, Rewal has employed prefabricated waffle slabs in conjunction with circular reinforced concrete columns to achieve a grid having a clear span of 10 metres by 10 metres. This sufficiently large clear span allows a free plan that accommodated varying floor area requirements of studios, production rooms, seminar halls, offices and classrooms. However, the structural competency of the CIET building is not flaunted by Rewal in its external appearance where local red sandstone remains at the forefront.

### 3.2.7. Cultural Appropriateness

The courtyard in the CIET building not only acts as a climatological device but also acts as a focal point for communal interaction between the users of the building (Figure 12). The in-built steps in the courtyard invite users walking in the surrounding corridors to gather and sit in the courtyard [21] (p. 176). In addition to the courtyard, Rewal has provided roof terraces on the uppermost level. Traditionally, roof terraces in India have been used for outdoor sleeping on hot summer nights. In the CIET building, roof terraces have been provided for enjoying the mild summer evening breeze with views of the surrounding national park and the distant Qutab Minar [26]. The roof terraces serve an additional purpose as they allow users to enjoy the winter sun.

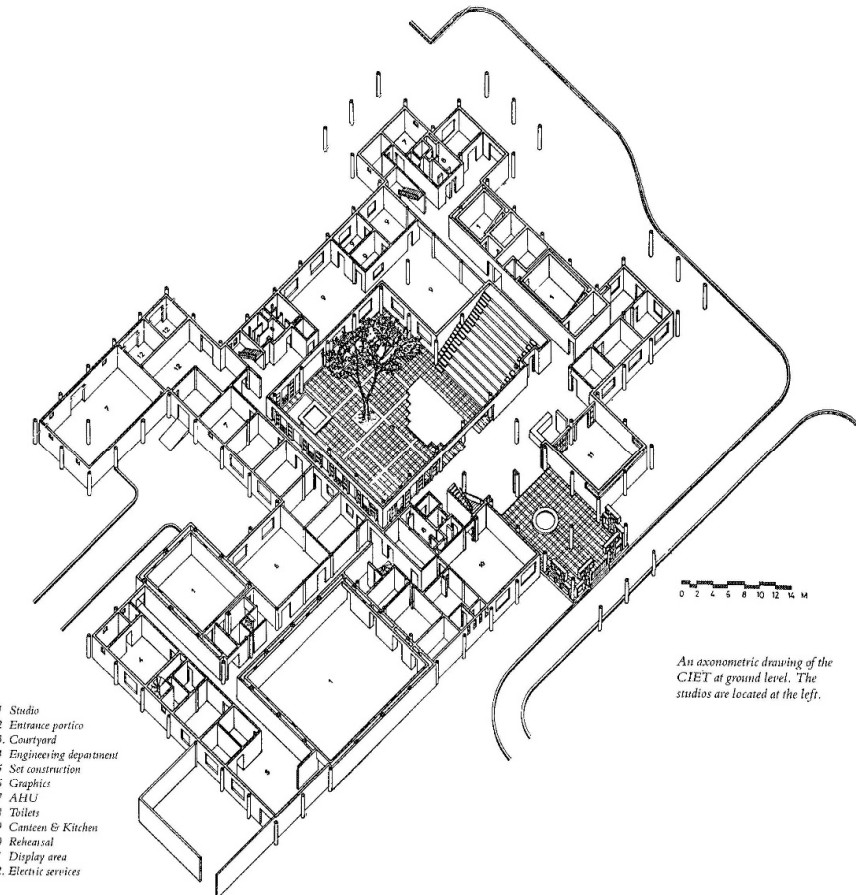

1 Studio
2 Entrance portico
3. Courtyard
4 Engineering department
5 Set construction
6 Graphics
7 AHU
8 Toilets
9 Canteen & Kitchen
10 Rehearsal
11 Display area
12. Electric services

*An axonometric drawing of the CIET at ground level. The studios are located at the left.*

**Figure 12.** Central courtyard acts as a focal point in the CIET building (Courtesy: Raj Rewal Associates).

### 3.3. India Habitat Centre (1993)

The India Habitat Centre (IHC) is a large complex that accommodates 30 to 40 institutions dealing with diverse issues related to habitat and environment (Figure 13). The 9-acre campus of IHC comprises of office workspaces, conference facilities, library, exhibition areas, auditorium and a variety of restaurants. Through this project architect Joseph Allen Stein has demonstrated that it is possible to attain humanist values even in large-scale projects having densities over 1000 persons per acre. Stein's upbringing and architectural education in the midwestern United States followed by more than thirty years of practice in Delhi has combined in his design a response to the local with certain universality.

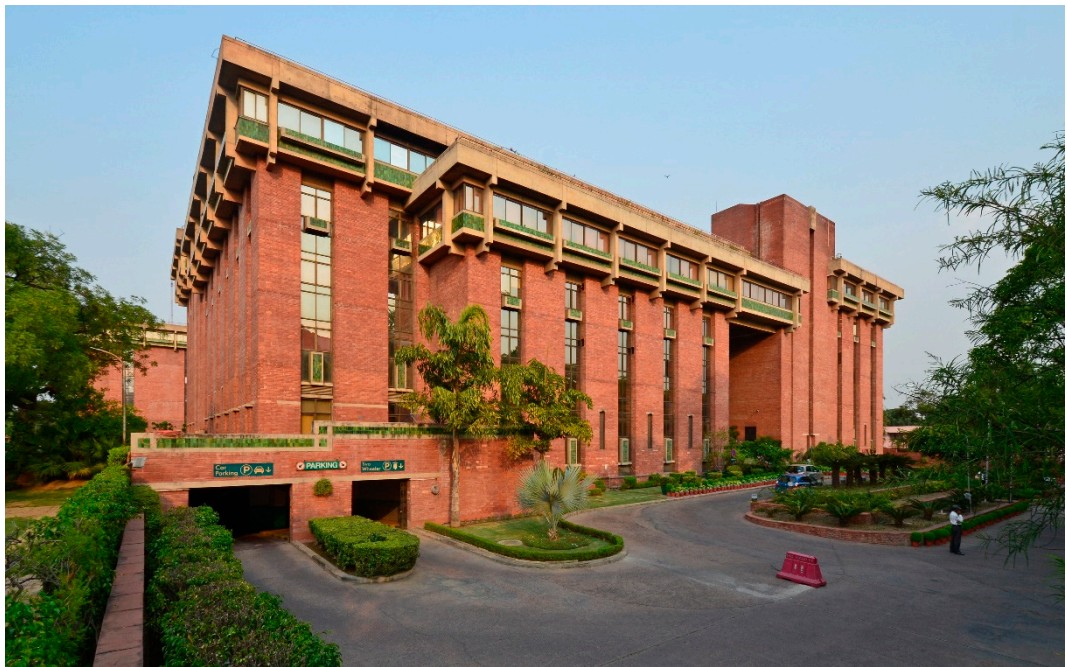

**Figure 13.** India Habitat Centre, New Delhi (1993) by Joseph Allen Stein (Photo: Author).

#### 3.3.1. Contextual Response

In comparison to the other buildings in the Lodi Estate, the IHC building is much larger in scale and has a very different character due to its unique programme (Figure 14). Despite the larger scale, the IHC building shows respect for the character of the surroundings by foregoing overindulgence. In its simplicity, the building's demeanour shows humility while seeking beauty without ostentation.

The larger scale of the IHC building also prompted Stein to transform his design approach from 'building in a garden', that he mastered in the nearby India International Centre (1962), to a new approach of 'garden in a building' [28]. Shaded gardens at ground level between the building blocks act as an oasis of quietness and greenery amidst the traffic and bustle of the city. The introverted nature of the IHC building is further reflected in the fenestration designs as the outer façades have narrow vertical slits while the inner façades have wide horizontal ribbon windows. Despite its introversion to the chaos of the city, the IHC complex is accessible on foot from all sides. Free pedestrian access along with interconnected garden courtyards create an open pedestrian continuum that helps linking and unifying the neighbourhood.

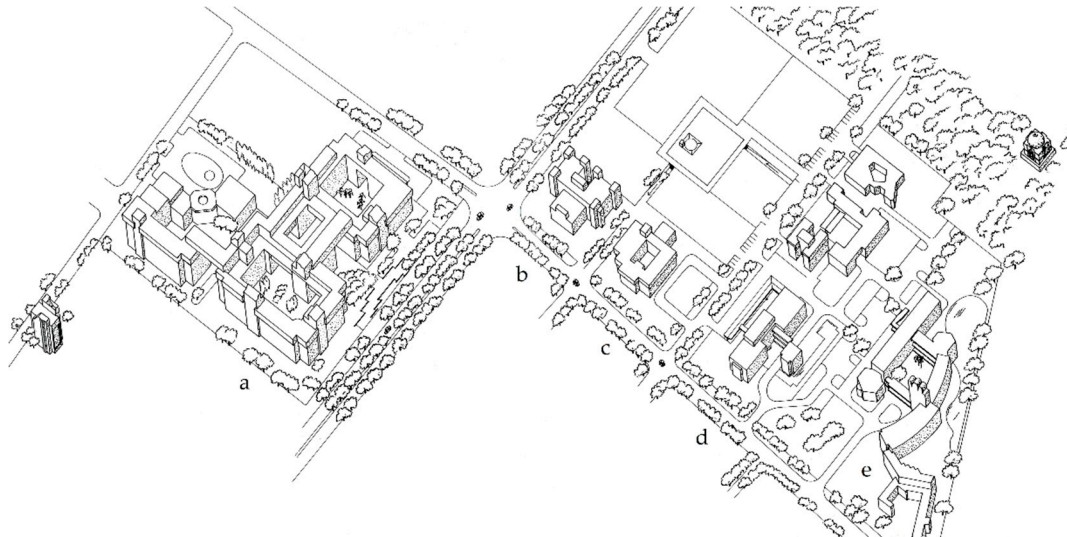

**Figure 14.** Isometric view of Lodi Estate comprising of the (**a**) India Habitat Centre; (**b**) WWF; (**c**) UNICEF; (**d**) Ford Foundation; and (**e**) India International Centre buildings [29] (p. 114).

### 3.3.2. Historical Knowledge

Stein's understanding of India's traditional architecture was not merely skin-deep. He recognised the continuing feature that linked the architecture of the region from traditional forms to recent works was the seeking of a space in shade. In particular, he was influenced by the shade-giving elements and spatial arrangements in the imperial palace at Fatehpur Sikri [29] (p. 131). The sixteenth-century palace complex integrates many shade-giving devices like *jaalis*, *chaajjas* and pavilions, intertwined with gardens, courtyards, pools and trees. The various monuments, palaces, assembly halls and public offices in the complex are laid on multiple axes and are interlinked by the shaded courtyards (Figure 15). The design of IHC also makes use of similar-sized internal courtyards and garden courts that are shaded by multiple devices and are interconnected to form a continuous public space. The overhangs at Fatehpur Sikri have been reinterpreted in the IHC building by projecting out the top two floors, while overhead screens above the courtyards further protect the Centre from excesses of tropical sunshine. Thus, by embodying learnings from the past, the IHC is able to offer a high quality of repose.

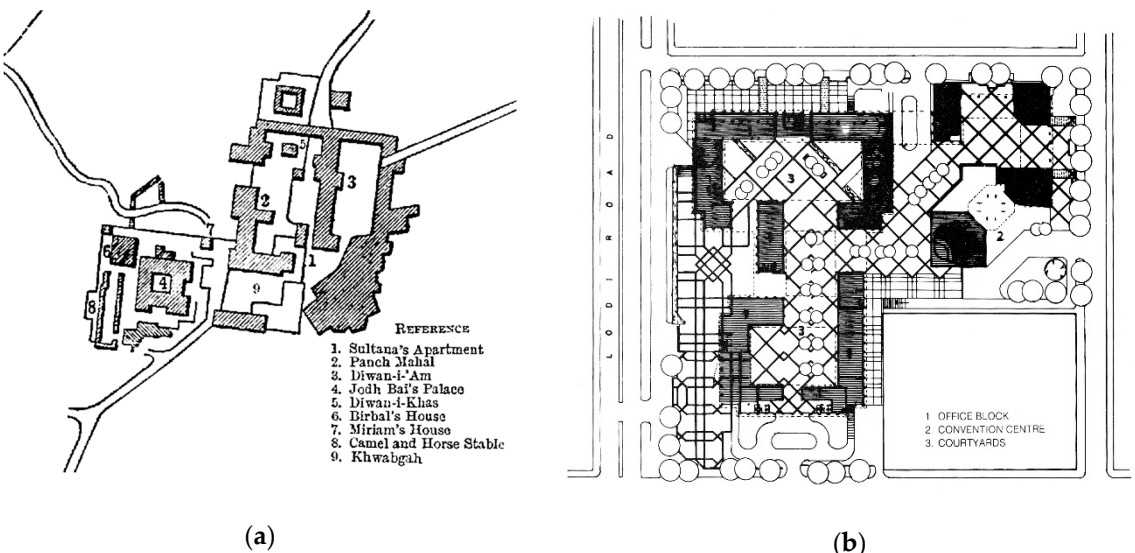

(**a**)　　　　　　　　　　　　　　　　　　　　　　　(**b**)

**Figure 15.** Interlinked shaded courtyards in the site plans of: (**a**) Fatehpur Sikri Palace [30] and (**b**) India Habitat Centre [31].

### 3.3.3. Climate Responsiveness

The design of IHC, to a great extent, relies on non-mechanical means for climate control. In order to reduce solar heat gain, the building volumes are organised around shaded courtyards. These courtyards are covered by sun-screen pergolas suspended from a space frame structure (Figure 16). The sun-screen pergolas contain angled panels designed to block the summer sun while letting in the winter sun [32]. As a result, the courtyards have acquired a unique microclimate which is conducive for repose. Moreover, the pergolas shade not only the courtyards below but also the inner façades of the building volumes.

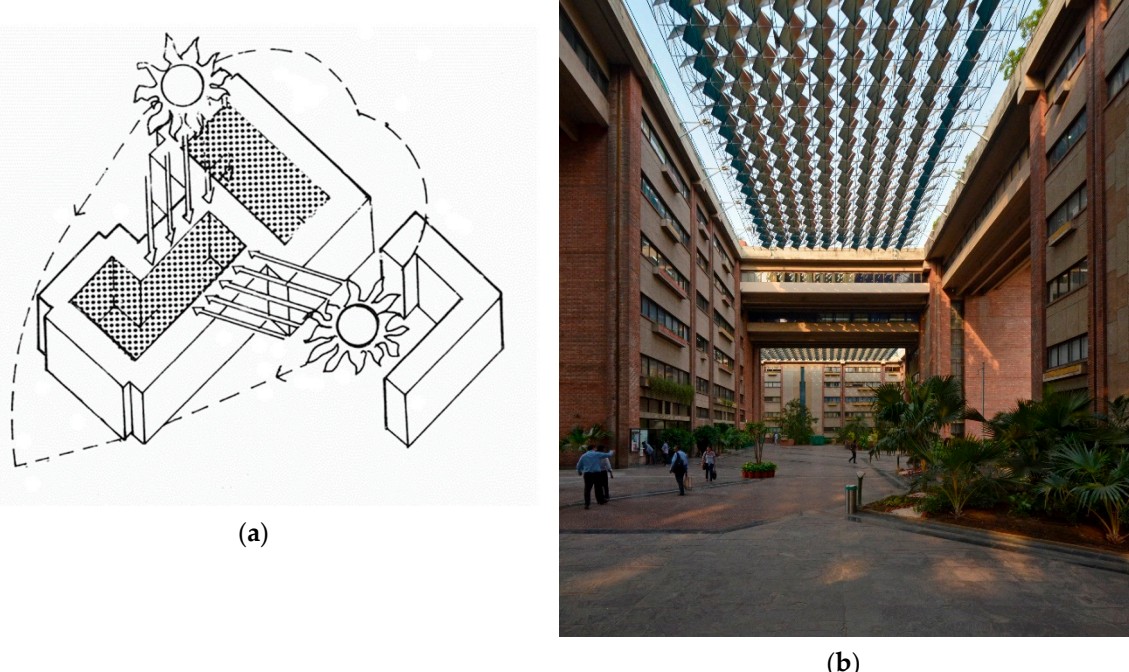

(**a**)

(**b**)

**Figure 16.** (**a**) Isometric drawing depicting variable shading of internal courtyards by sun-screen pergolas during summers and winters [29] (p. 131); (**b**) photograph of an internal courtyard of the India Habitat Centre (IHC) building with an overhead sun-screen pergola (Photo: Author).

Other passive climate-control measures include projecting out the top two floors for self-shading, use of cavity walls and a controlled use of glass. To ensure adequate daylighting, the floor plates have been limited to a width of 15 metres so that no individual is more than 7.5 metres away from a window. However, the climate responsiveness of the IHC building could have been further improved had the fenestrations on each of the outer façades been designed differently as per their specific solar orientation.

### 3.3.4. Ecological Sensitiveness

Stein often expressed his concern about the negative impact on earth's ecology from the ever-increasing human population [29] (p. 53). Although the IHC had to accommodate a high density of approximately 1000 persons per acre, Stein strove to make amends for the ecological impact of the building's construction. First and foremost, Stein has tried to recover the lost green cover on the ground by creating roof terrace gardens. Moreover, an abundant use of planter boxes has been made throughout the complex in an attempt to introduce green wherever possible. On the ground level, a number of garden courts have been created between the building volumes and above the basement car parking (Figure 17).

Furthermore, Stein has introduced the concept of vertical hanging gardens in the design of IHC. Planter boxes have been integrated with the vertical faces of building volumes, which are additionally

supplemented by a watering system [31]. Collectively, all these measures are able to alleviate the building's ecological impact.

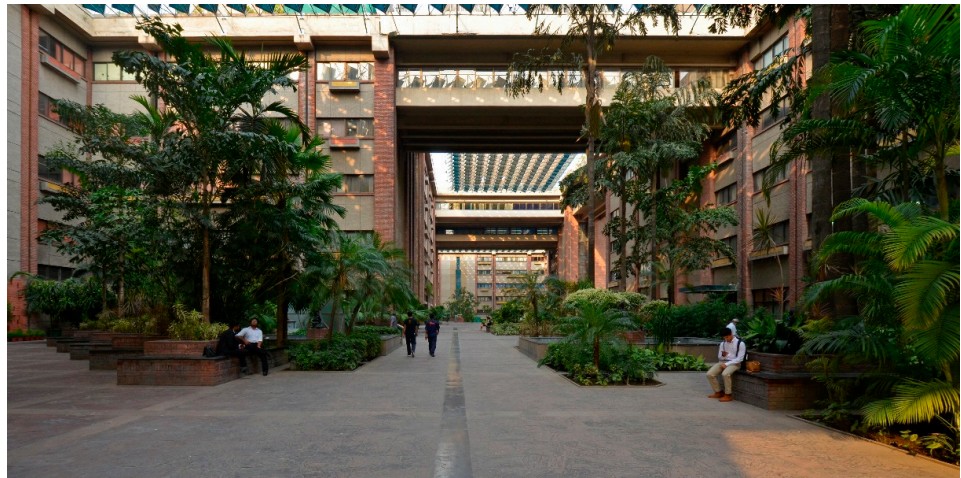

**Figure 17.** Garden courts between building volumes of the India Habitat Centre (Photo: Author).

### 3.3.5. Local Materials and Construction

The structural frame of IHC has been fabricated from reinforced cement concrete cast in situ. Popularised by Le Corbusier in India, in situ concrete takes advantage of the availability of relatively cheap manual labour in the country. In his earlier works realised in Delhi, Stein had left the concrete structural frames exposed. However, by the 1980s, Stein had become dissatisfied with the quality of exposed concrete often achieved by local contractors [29] (p. 126). Therefore, Stein chose a continuous cladding system for the IHC building instead.

Stein had originally envisioned the cladding in brushed stone aggregate plaster, but had to later switch to brick as the client felt that brick better symbolised 'habitat', which was central to the theme of the Centre [21] (p. 213). Local kilns were approached to manufacture special Lakhori bricks—a thin, flat variety popular during the Mughal era (Figure 18). Variegated courses of Lakhori bricks envelop the building's outer façades while brushed stone aggregate plaster has not been completely discarded as it clads the building's inner façades.

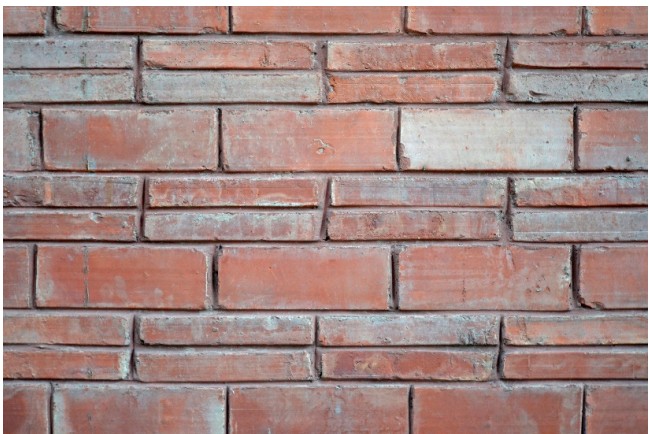

**Figure 18.** Special Lakhori bricks used for cladding the IHC building (Photo: Author).

### 3.3.6. Technological Sustainability

In order to effectively shade the courtyards between the building volumes, Stein needed overhead screens to cover them up. Covering the courtyards was, however, by no means an easy task as the overhead screens were required to span a width of 30 metres. As a consequence, Stein developed three

large sunscreen pergolas, one for each courtyard, made up of hundreds of panels suspended from large space frames (Figure 19). Made from sailcloth, these panels are dexterously angled to block the summer sun while letting in the winter sun.

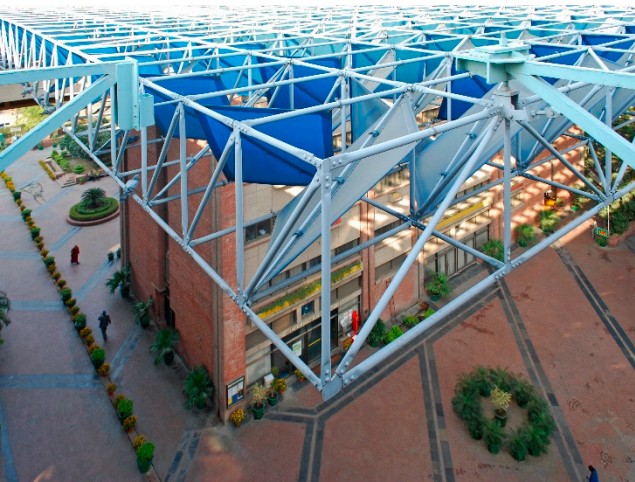

**Figure 19.** Sailcloth panels suspended from space frame to shade courtyards below (Photo: Author).

The innovation of sunscreen pergola significantly improves the enclosure qualities of the courtyards [21] (p. 213). However, instead of fetishizing his innovation, Stein has used light blue coloured panels and space frames to blend them with the sky up above.

### 3.3.7. Cultural Appropriateness

The IHC complex hosts a variety of public and private sector enterprises under one roof. In this light, the complex has been designed in a way so as to encourage easy interchange of information amongst professionals working in diverse habitat-related fields. The building volumes are organised around shaded atrium-like courtyards that are interlinked to form a series of pleasant civic spaces. Seatwalls are spread throughout these courtyards to further accentuate the communal setting (Figure 20) [33]. Moreover, to evade the tyranny of motor vehicles, all traffic has been directed into two levels of basement parking. Thus, the IHC complex is able to provide proximity to different institutions facilitating a synergetic interrelationship between them.

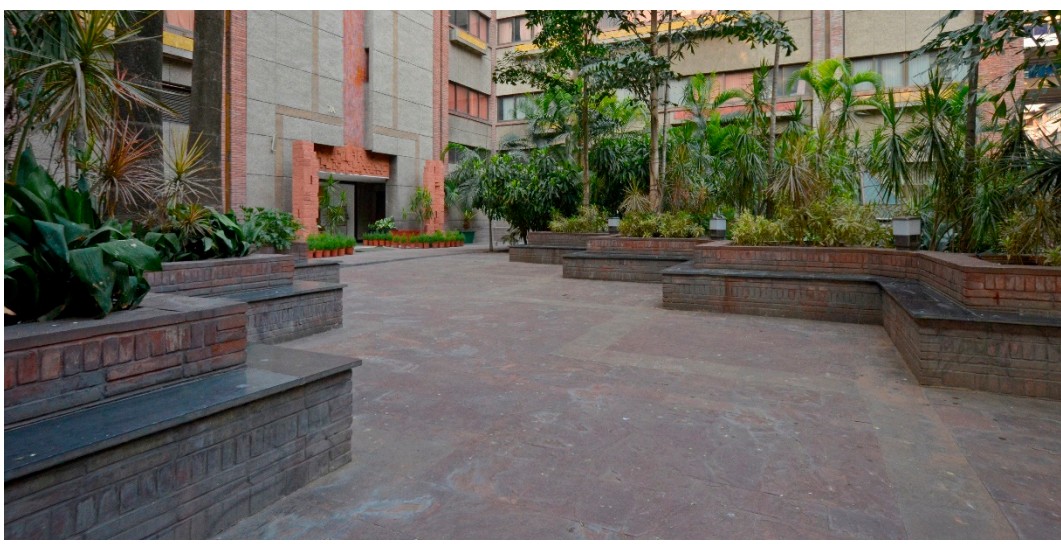

**Figure 20.** Seatwalls provided in the courtyards of India Habitat Centre (Photo: Author).

### 3.4. Development Alternatives Headquarters (2008)

Development Alternatives (DA) is a non-governmental organisation established in 1983 to promote environmentally-appropriate technology that fosters socioeconomic development in India. The six-storey headquarter building of DA provides 4500 square metres of usable area for the 200 employees of the organization (Figure 21). Through his design architect Ashok B Lall demonstrates how conventional parameters of sustainable architecture can be fulfilled while relying on low-cost solutions based on fundamental lessons in local tradition. Lall's education at the Architectural Association School in London exposed him to Western building techniques and methodology, while his upbringing and more than twenty-five years of practice in Delhi has sensitised his work to locally rooted traditions.

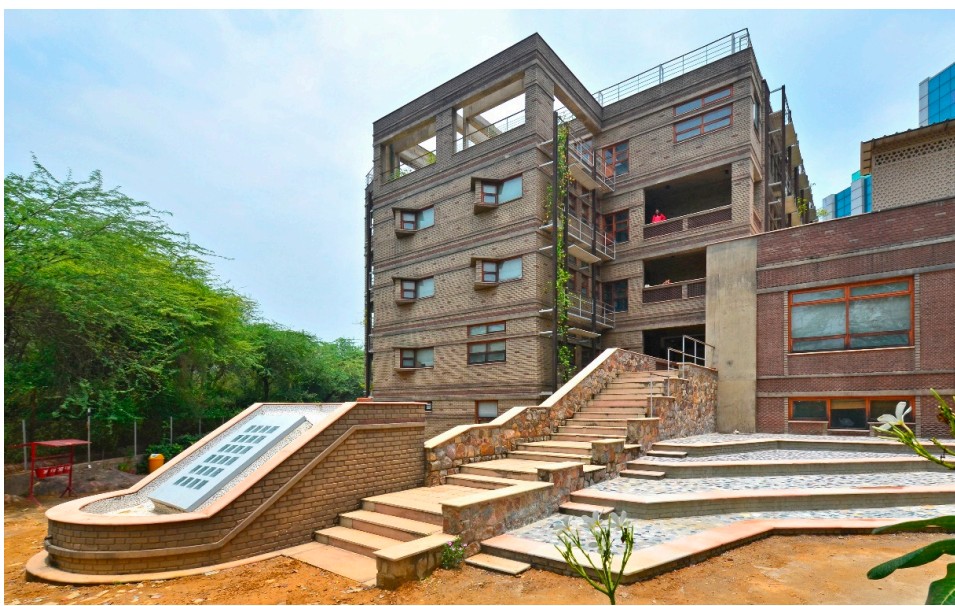

**Figure 21.** Development Alternatives Headquarters, New Delhi (2008) by Ashok B Lall (Photo: Author).

3.4.1. Contextual Response

The DA building is located on the south-western corner of the Qutab Institutional Area in South Delhi (Figure 22). The precinct of this institutional area largely consists of sizable free-standing office buildings that symbolise corporate power through their architecture (Figure 23). Being on the corner of the institutional area, the DA site is also bound by a dense forest reserve on two of its sides.

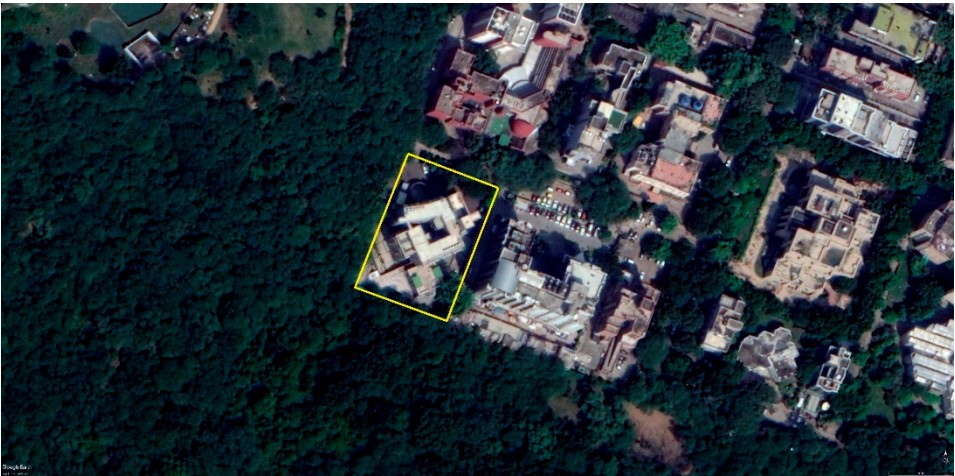

**Figure 22.** Satellite image of the Development Alternatives (DA) building located on the south-western corner of Qutab Institutional Area (Image courtesy of Google Earth).

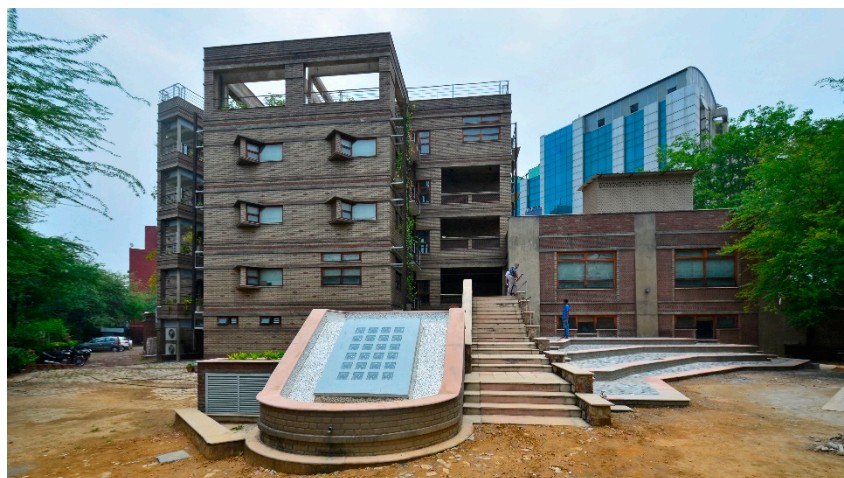

**Figure 23.** Earthy finish of the DA building in contrast with the neighbouring curtain-glass buildings visible in the background (Photo: Author).

Rather than responding to the corporate architecture of the adjacent buildings, Lall instead chooses to address mainly two determinants: The old DA building and the adjoining forest. Completed in 1988 and situated on the same site, the old DA building had to be replaced as it had outgrown the functional requirements of the organisation. During consultations with the DA staff in the initial stages of the project, the architect discovered that it was crucial to the DA staff that the new building evoked the memory of the old one [34]. Therefore, the new building uses elements, forms and materials that recall those of the old building: Domes, vaulted ceilings and compressed-earth blocks.

The DA building's relationship with the adjoining forest is not very strong but it does take care of views towards the forest in its window designs. Moreover, the earthy material palette of the DA building does not contrast with the forest as much as the neighbouring curtain glass buildings (Figure 23).

3.4.2. Historical Knowledge

Even though the primary focus of the DA building was to fulfil the parameters of sustainable architecture [35], Lall has demonstrated that it is possible to achieve sustainability goals without breaking away from the past. The DA building in its design interprets traditional building elements such as domes, courtyards, verandas, terraces, balconies and screens. Shallow domed ceilings in brick span many of the spaces instead of flat reinforced concrete ceilings that are usually employed in modern constructions of similar scale (Figure 24).

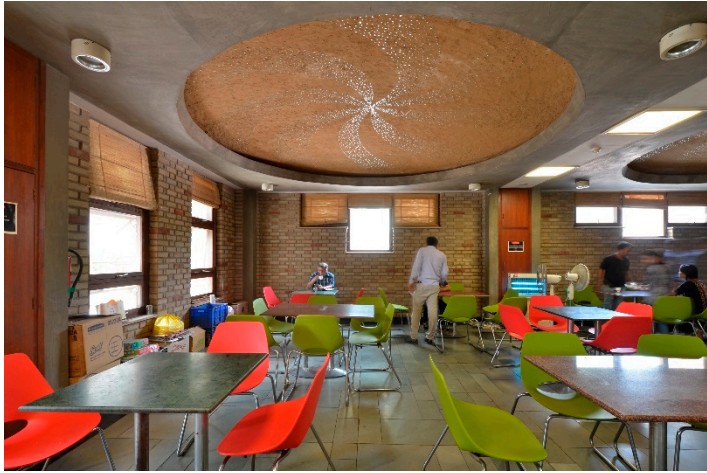

**Figure 24.** Shallow domed ceilings span the interior spaces of the DA building (Photo: Author).

Despite the constraints of a restricted plot size, the architect has managed to provide a narrow central courtyard (Figure 25). The courtyard is helpful in visually connecting the different offices and meeting rooms located across different floors of the building.

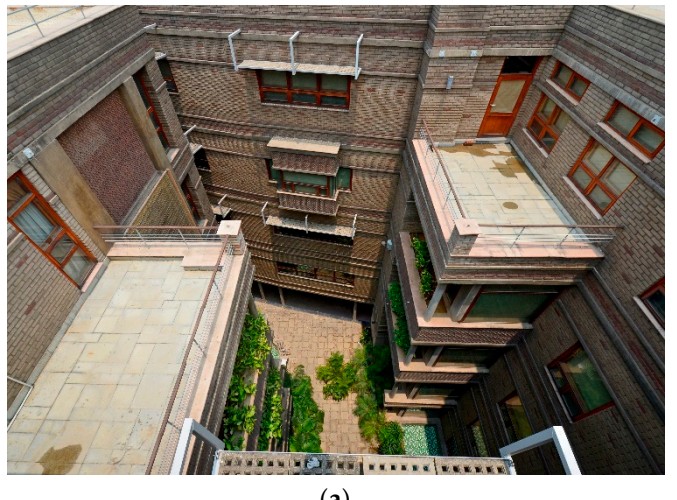 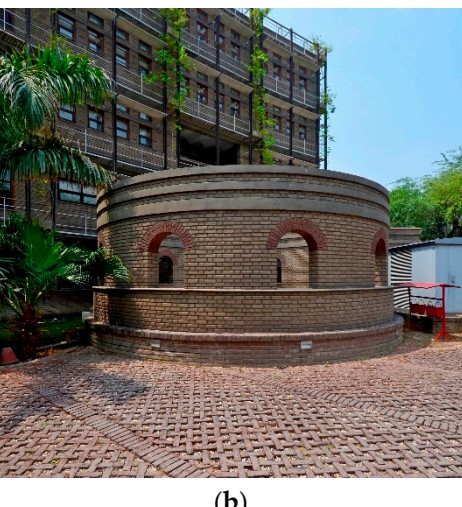

| (**a**) | (**b**) |

**Figure 25.** Traditional building elements such as the (**a**) courtyard and (**b**) *baoli* reinterpreted in the DA building (Photos: Author).

Another device in the DA building seeking a connection with the region's traditional architecture is the *baoli*. *Baolis* are subterranean stepwells unique to the Indian subcontinent [36]. Although *baolis* come in a variety of shapes, sizes and styles, the modernised *baoli* in the DA building is drum-shaped having arched clerestory windows (Figure 25). Located on the northern corner of the plot and accessible only from the basement level, the *baoli*, however, seems neglected in the overall scheme of the building.

### 3.4.3. Climate Responsiveness

Lall has taken a string of measures to ensure that the DA building responds diligently to the climate of Delhi. To ensure better thermal performance, masonry cavity walls are used on the exteriors and less than twenty percent of the building envelope consists of glazing. Fenestrations on each façade are designed differently as per their solar orientation (Figure 26). Balconies provided on the south-east and north-west façades further help in shading the windows below them. In addition, the surface of the roof is finished in white mosaic flooring so as to reflect as much of the sun's energy as possible.

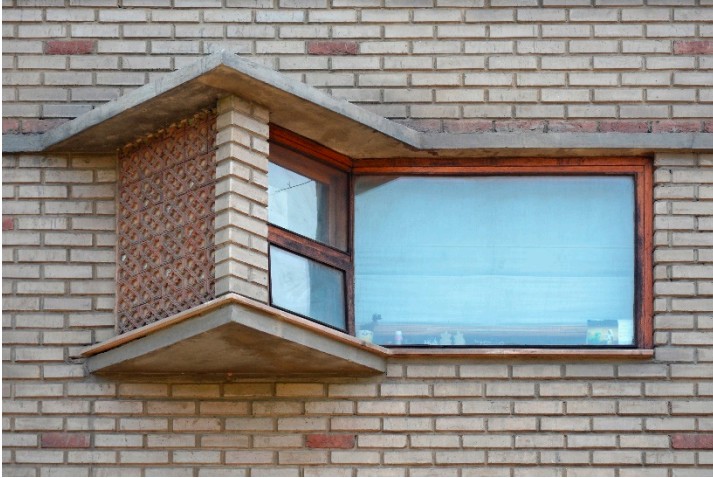

**Figure 26.** Window on the west façade having a prism-shaped protrusion to block the afternoon summer sun (Photo: Author).

The DA building employs a hybrid air-conditioning system that uses both evaporative cooling and refrigerant-based cooling so as to reduce the energy consumption by thirty percent [37] (p. 12). Moreover, the DA building maintains an indoor maximum temperature of 28 °C at 60% relative humidity, which is higher than the norm of 24 °C temperature maintained in corporate buildings. The provision of ceiling fans and a liberal dress code for DA employees ensure adequate thermal comfort in the DA building even in 28 °C (Figure 29). Operational costs are further reduced by the building's narrow floor plates as the workspaces are comfortably illuminated by daylighting.

### 3.4.4. Ecological Sensitiveness

In the DA building, Lall has prioritised the use of recycled materials over virgin materials. Earthen blocks recovered from the demolished old DA building were recycled into new compressed-earth blocks for the new building. The building also uses industrial wastes such as fly-ash for blocks used in interior and exterior walls, waste expanded polystyrene to insulate cavity walls and broken white ceramic tiles for mosaic flooring on the roof top. Although the building does not use recycled timber for woodwork and furniture, the timber used is sourced from certified managed plantations [37] (p. 40).

A unique feature in the external walls of the DA building is the insetting of clay pots (Figure 27). Inserted randomly into the external walls, these clay pots offer nesting places for birds, bees and squirrels from the adjoining forest.

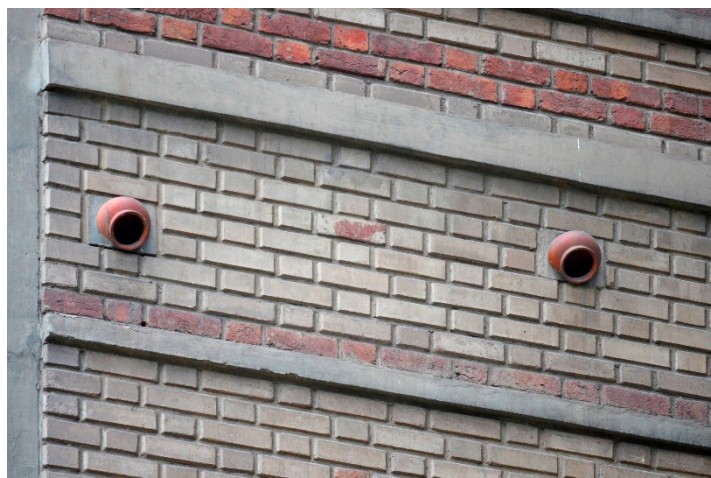

**Figure 27.** Clay pots inserted randomly into the external walls of DA building offer nesting places for birds and bees (Photo: Author).

All rainwater that falls on the DA site is collected for recharging the ground aquifer. In addition, the wastewater from the DA building is treated, filtered and reused for watering plants and flushing toilets.

### 3.4.5. Local Materials and Construction

The predominant materials used in the DA building are compressed-earth blocks (Figure 28) and fly-ash lime gypsum blocks. Although these materials are relatively new to the region, they have been chosen for their low cost and eco-friendly properties. The compressed-earth blocks have been recycled from the earthen blocks of the old DA building while fly-ash blocks have been produced from fly-ash sourced from a local power plant. For the purpose of flooring, unpolished granite and sandstone quarried in north India have been used.

Although the DA building is a complex ensemble of a variety of materials and forms, the elements comprising the whole have been kept simple. As a result, a variety of local craftsmen, small-scale contractors and daily wagers had the opportunity to be involved in the construction process. Plastering of the shallow domes with dung (Figure 24), filling of terracotta *jaali* screens with vermiculite plaster

(Figure 28), teak wood carpentry and laying of brick pavers are some of the construction processes that benefited from the abilities of the local workforce.

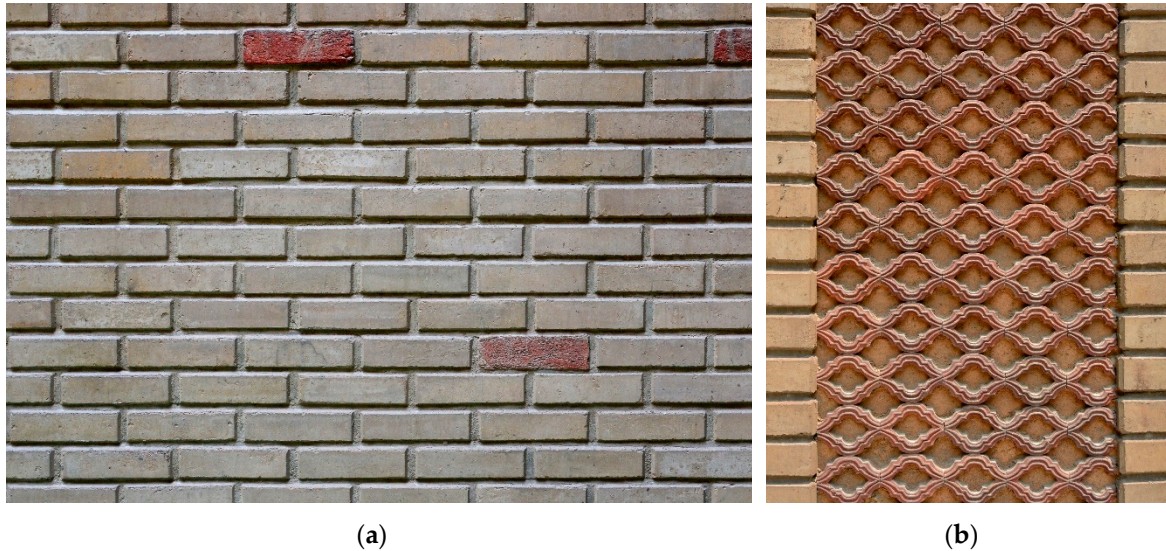

(**a**)    (**b**)

**Figure 28.** The material palette of the DA building consists of earthy materials such as (**a**) cement-stabilised compressed-earth blocks and (**b**) terracotta tiles (Photos: Author).

### 3.4.6. Technological Sustainability

In the DA building, Lall has managed to find appropriate combinations of using modern materials like steel and glass with natural materials like mud and wood (Figure 29). This hybrid character has evolved despite the fact that Lall has sparingly used materials like glass and steel that possess high embodied energy. Glass and steel have been used efficiently and only where their use is pertinent.

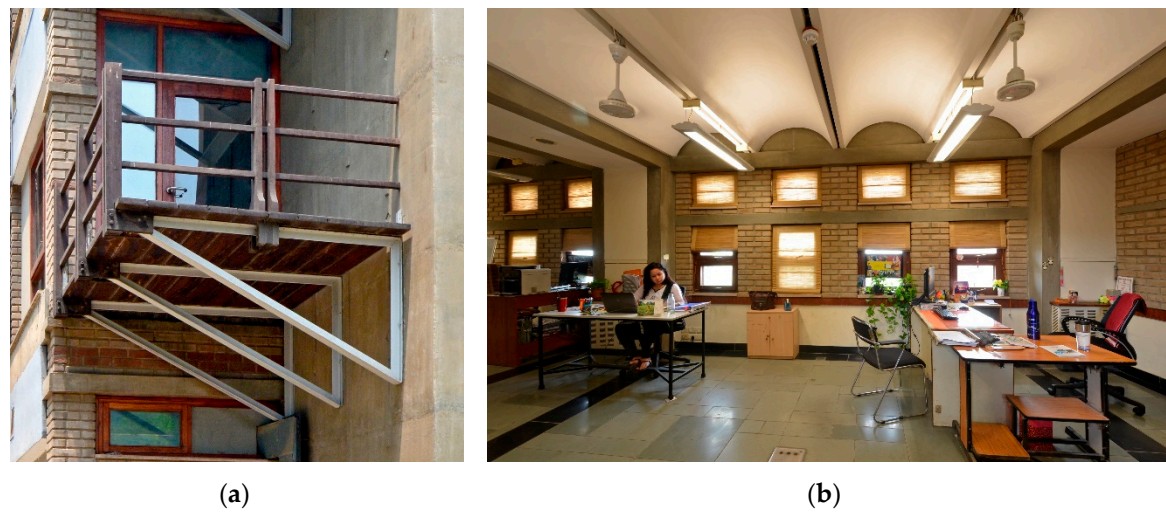

(**a**)    (**b**)

**Figure 29.** (**a**) Combination of natural and modern materials forming a balcony in the DA building; (**b**) intimate workspaces in the DA building (Photos: Author).

Lall has even pushed the boundaries of existing technology in adopting a hybrid air-conditioning system that uses both evaporative cooling and refrigerant-based cooling to reduce energy consumption by thirty percent. Another innovation in the DA building is the use of vaulted precast-concrete deck elements for spanning most spaces (Figure 29). Use of vaulted ceilings resulted in reduction of steel consumption by half in comparison with conventional structures of comparable size.

The DA building's appearance, however, masks its technological innovations and emphasises on the traditional wisdom inherent in it. More importantly, the building demonstrates that a green building need not necessarily be a high-technology-centred creation but can rather develop from the heritage of ideas of the place.

### 3.4.7. Cultural Appropriateness

The design process for the DA building involved a highly participatory process as views of the client and users were taken at various stages of design development [37] (p. 14). These structured consultations helped in building a consensus as to how the DA building would fulfil the needs and aspirations of all user groups. A notable outcome of this process is the privacy afforded to workspaces in the building (Figure 29). Unlike conventional office buildings, the DA building avoids having large open-plan workspaces. The workspaces instead have a degree of privacy and intimacy resulting in an almost domestic scale to the office environment. Small separate balconies and verandas (on ground level) attached to the workspaces further add to this sense of domesticity [34].

In addition to balconies and verandas, a variety of other informal spaces like cafeteria, courtyard, amphitheatre and *baoli* have been included to serve the anthropological needs of the users. Furthermore, the choice of natural materials lends a sense of solidity to the building while adding to the design's tactile experience.

### 3.5. Dilli Haat Janakpuri (2014)

Dilli Haat Janakpuri has been modelled as an urban *bazaar* that accommodates an elaborate programme consisting of craft shops, a music museum, workshops, an exposition hall, an auditorium, an open-air amphitheatre and a large food court (Figure 30). Commissioned by the Delhi Tourism and Transportation Development Corporation (DTTDC), the six-acre complex of Dilli Haat provides a seamless link between artisans and consumers to transact. Architect Sourabh Gupta of Archohm Consults in his design manages to maintain a certain tension between modern and traditional, and universal and local. Gupta's architectural education at the Centre for Environmental Planning and Technology (CEPT) in Ahmedabad and Technical University Delft in the Netherlands has enabled him to blend the local and the universal while avoiding the limitations of each.

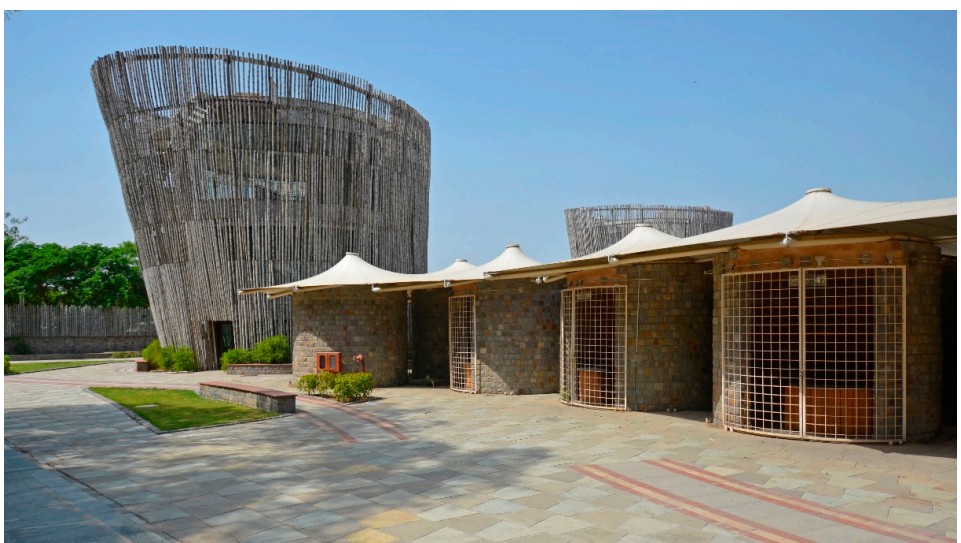

**Figure 30.** Dilli Haat Janakpuri, New Delhi (2014) by Archohm Consults (Photo: author).

### 3.5.1. Contextual Response

The six-acre plot of Dilli Haat is located in a complex urban milieu of western Delhi's Janakpuri district (Figure 31). The plot is bound by the Tihar prison complex, a bus depot, a huge green belt

and a densely populated middle-class residential colony. Despite a diverse mix of urban development in Janakpuri, the overall landscape of the area is banal and infamous for the country's largest prison Tihar Jail. Therefore, the design of Dilli Haat consciously overlooks the chaotic urban context of the prison complex and the haphazard residential neighbourhood, and instead chooses to focus on the surrounding green belt (Figure 32). The green belt is made up of tall trees of mainly of the Eucalyptus variety. Sourabh Gupta has designed the Dilli Haat as an extension of this green belt. The existing trees in the site have been retained and tall basket-like structures clad with bamboo dominate the skyline of Dilli Haat. In addition, much of the built-up area in the site is covered with green roofs or creepers resulting in the continuation of the green cover.

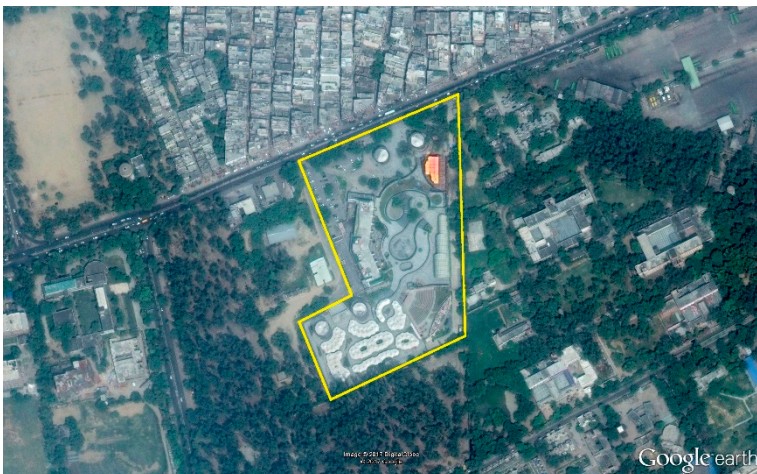

**Figure 31.** Satellite image of Dilli Haat Janakpuri (Image courtesy of Google Earth).

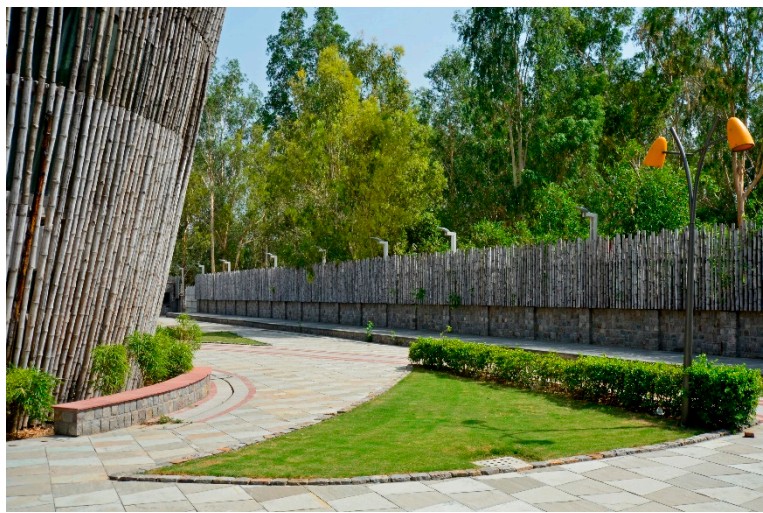

**Figure 32.** Tall bamboo-clad structures respond to the green belt surrounding the Dilli Haat site (Photo: Author).

### 3.5.2. Historical Knowledge

A *haat bazaar* is a traditional trading venue for local people in villages and small-towns of South Asia. Held on a regular basis, *haat bazaars* enable craftsmen and artisans to sell their handicraft products directly to their consumers. From the conception of the project itself, the Dilli Haat Janakpuri was required to reinterpret and recreate these *haat bazaars*.

The informal zone of the Dilli Haat, consisting of 100 craft shops, interprets the traditional spatial arrangement of Indian *bazaars* (Figure 33). Craft shops are arranged in clusters of 5–6 shops and these clusters are further composed in an organic layout to form a *bazaar*. Paved allies meander along the

craft shop clusters and small green patches lending an intimate character to this zone (Figure 37). However, having to accommodate a wide range of functions mandated by the elaborate design brief of the project, the overall design of Dilli Haat ends up diluting the spirit of a crafts *bazaar* [38].

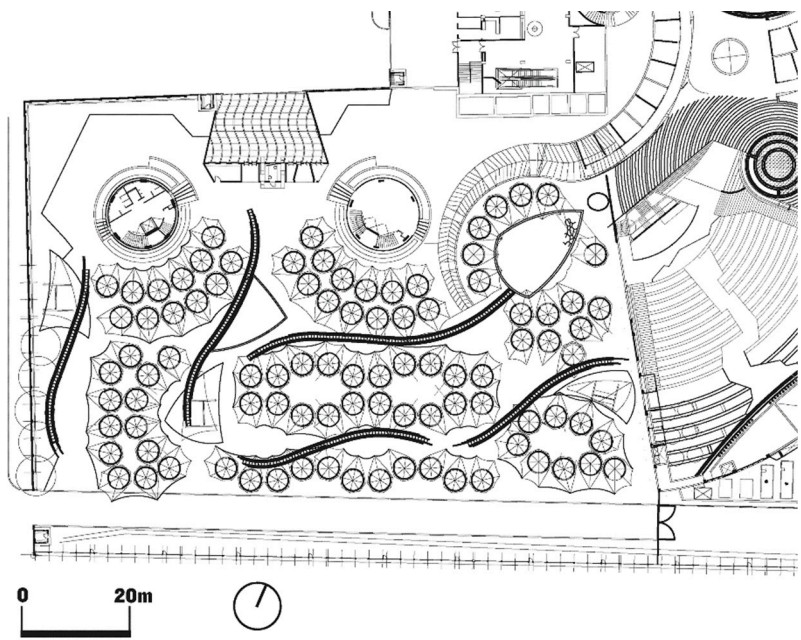

**Figure 33.** Floor plan of the informal zone having craft shops (Courtesy: Archohm Consults).

### 3.5.3. Climate Responsiveness

The built forms in the Dilli Haat have been clad in stone that delays heat from reaching the interiors due to its thickness and thermal mass [39]. The craft shops are covered with canopies of translucent tensile membrane allowing toned daylight into the shops. Floating above the circular walls of the craft shops, these canopies also enhance the air circulation within. The large overhang of the tensile canopies shades the craft shops against harsh sun, while the ceiling fans installed underneath the canopies further improve the thermal comfort in the shops (Figure 34).

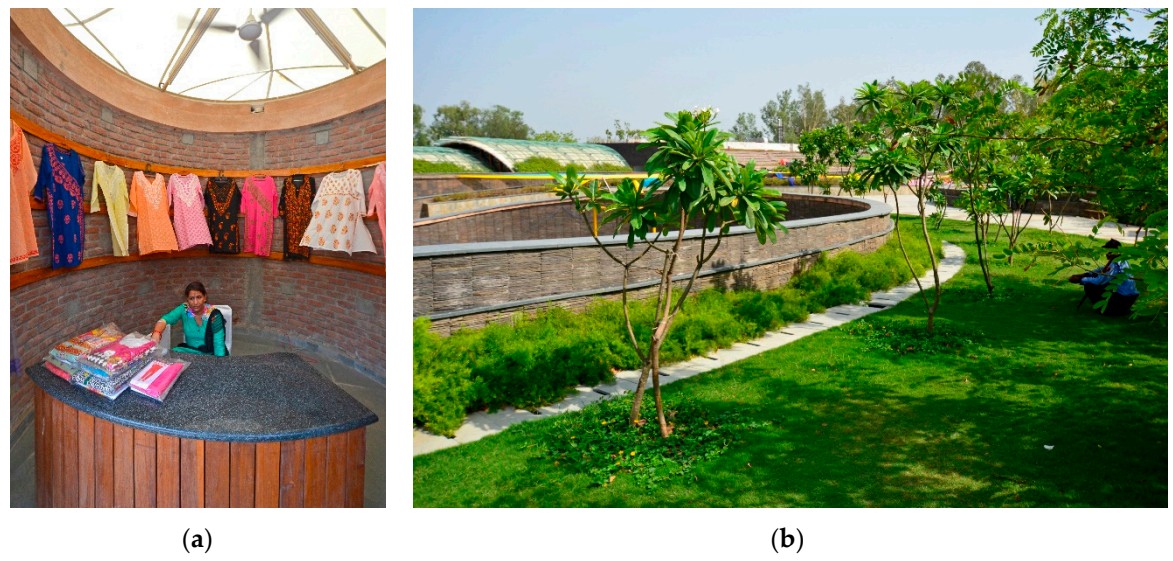

(**a**) (**b**)

**Figure 34.** (**a**) Tensile canopies with ceiling fans cover the craft shops; (**b**) large usable green roofs above air-conditioned shops (Photos: Author).

The 46 air-conditioned shops and the auditorium are covered with green roofs while the exposition hall is covered with creepers to help reduce the heat flux through the roofs (Figure 34).

### 3.5.4. Ecological Sensitiveness

Although the original site of Dilli Haat was more or less flat, it had plenty of green cover on it. The challenge for the architects therefore was to retain or replenish the green cover despite having to accommodate an elaborate building programme. Accordingly, the existing trees on the site have been retained in the layout and most of the built-up areas are covered with a layer of greenery to somewhat restore the initial green cover (Figure 34). Rainwater harvesting is another eco-sensitive feature incorporated in the site planning of Dilli Haat.

### 3.5.5. Local Materials and Construction

The idea of Dilli Haat Janakpuri was based on two other Dilli Haats developed by the same client in other parts of Delhi, one in 1994 and another in 2007. The material palette in both the earlier Dilli Haats is dominated by exposed brickwork. In order to create a unique identity for Dilli Haat Janakpuri, Sourabh Gupta had to opt for other locally-available materials besides brick. Therefore, different varieties of stone such as Delhi quartzite, Kota stone, red Agra sandstone and slate are used for cladding and paving (Figure 35).

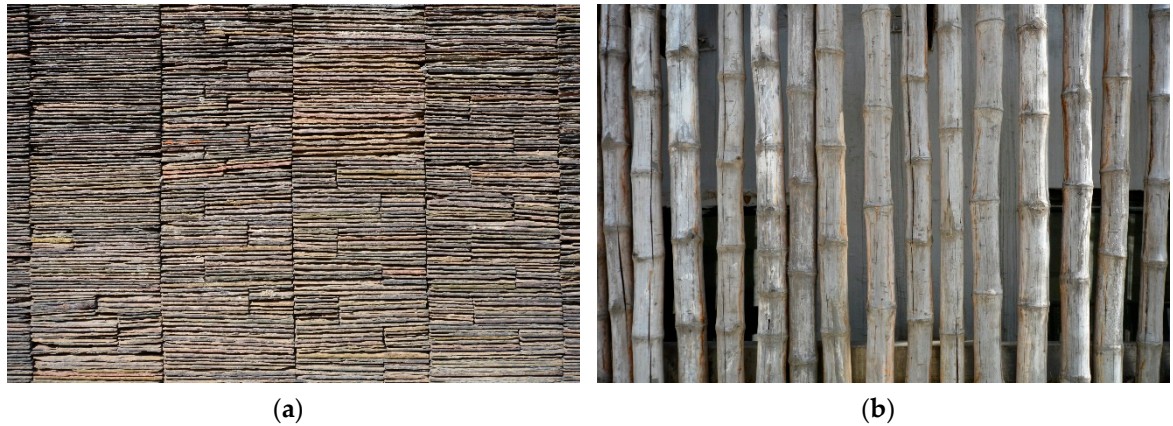

(**a**)                                              (**b**)

**Figure 35.** Cladding materials used in Dilli Haat Janakpuri: (**a**) Delhi Quartzite Stone and (**b**) bamboo (Photo: Author).

Another natural material used throughout the complex is bamboo. It is a renewable, environment-friendly material available abundantly in India. Besides screening the four 8-metre high towers, bamboo is also used for fencing the site boundary (Figure 32).

### 3.5.6. Technological Sustainability

In Dilli Haat Janakpuri, the architect has efficiently combined natural materials with industrial materials. The use of technologically-advance canopies above the craft shops is the centrepiece of Dilli Haat's design (Figure 36). A combination of steel and translucent tensile membrane allows the canopies to have large overhangs in addition to letting in diffused daylight into the craft shops. The architect, however, balances the use of technologically-advance canopies in the craft shops with walls clad in conventional natural stone masonry.

Another advanced material used in Dilli Haat is hollow structural steel tubes (HSS). HSS are used to fabricate linear vaults that span the 18-metre width of the large exposition hall. These steel vaults are not, however, flaunted in their external appearance as they are covered with creepers on the exterior instead. As evident, advanced technology is used in a functionally optimal and sustainable manner by the architect in the design of Dilli Haat Janakpuri.

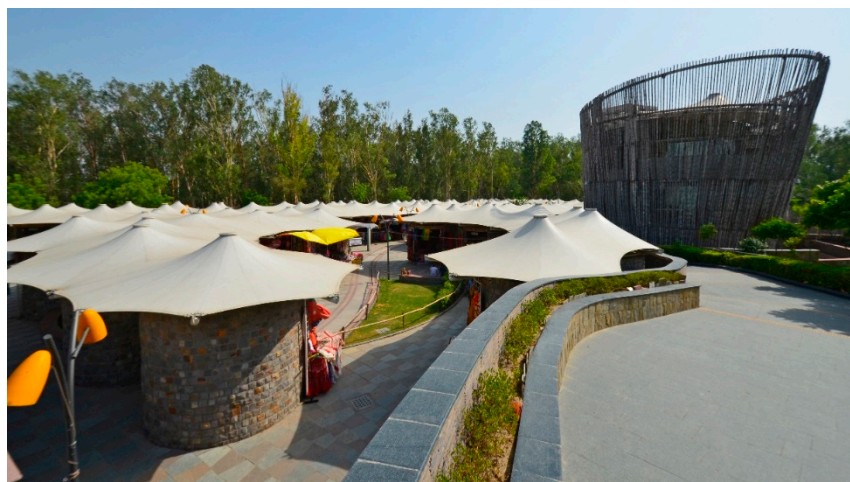

**Figure 36.** Tensile canopies covering the craft shops at Dilli Haat (Photo: Author).

3.5.7. Cultural Appropriateness

Being a cosmopolitan city, the demographic of Delhi consists of people belonging to a wide variety of economic and social backgrounds. In order to attract people from all strata of society, the design of Dilli Haat Janakpuri provides for outdoor craft shops, selling reasonably-priced goods, along with air-conditioned shops that cater to more upscale customers.

The built-forms consisting of shops, food court, exposition hall, etc. are organised around a variety of open spaces like courtyards, amphitheatre, paved allies and green patches (Figure 37). This has resulted in warm intimate spaces built to human scale [40].

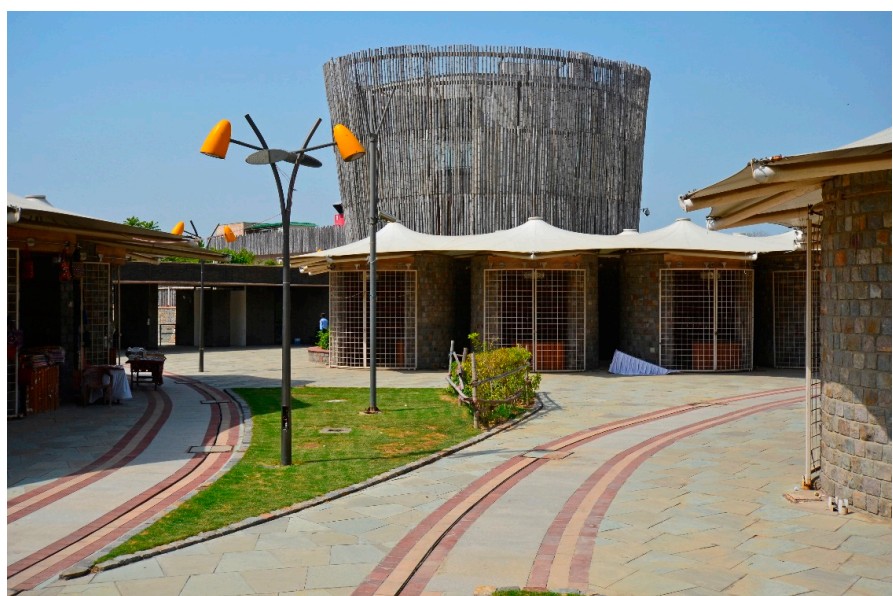

**Figure 37.** Built-forms organised around paved allies and green patches at Dilli Haat (Photo: Author).

The project also seems to fulfil the role of a neighbourhood park as residents of the nearby residential colonies frequent the complex, despite an entrance fee, to savour the warm atmosphere created by the green roofs, the amphitheatre and the various courtyards.

**4. Discussion**

Despite a common identification criterion, we can observe that the buildings differ substantially in their methods of fulfilling the fundamentals of critical regionalism (Table 1). Regionalist architects

have adopted different approaches to respond to the urban context of their respective sites. While the CIET building and the Dilli Haat choose to relate their built-forms to certain neighbouring elements such as the Qutab Minar and the woodland respectively, the DA building on the other hand recalls the old building that it had replaced. The IHC building on the other hand uses simplicity and humility in its expression out of respect for the surroundings.

**Table 1.** Comparison of different approaches to each aspect of critical regionalism.

| | Yamuna Apartments | CIET | India Habitat Centre | Development Alternatives | Dilli Haat Janakpuri |
|---|---|---|---|---|---|
| **Contextual Response** | Responds to generalised modernist architecture of urban India | Use of red sandstone relates to neighboring Qutab Minar | Foregoes over-indulgence to compensate for larger scale | Responds to the design of the old DA building located on the site | Bamboo-clad structures respond to the nearby woodland |
| **Historical Knowledge** | Organisation of housing blocks based on layout of typical Indian village | Central courtyard is reminiscent of a *madarsa* | Interlinked shaded courtyards influenced from Fatehpur Sikri palace | Interprets traditional elements such as domes, courtyards, verandas, terraces, balconies and screens | Organic layout of craft shops interprets traditional spatial arrangement of Indian *bazaars* |
| **Climate Responsiveness** | Self-shading from deep balconies; well-distributed openings for cross-ventilation | Wide corridors and deep balconies create buffer from sun | Large sun-screen pergolas shade the building volumes and courtyards | Fenestrations on each façade designed differently as per solar orientation | Roof canopies shade against harsh sun while allowing air circulation |
| **Ecological Sensitiveness** | Vegetation cover provided along the internal streets | Courtyard created around a pre-existing tree to preserve it | Roof gardens, vertical gardens, planter boxes and garden courts recover lost green cover | Use of recycled materials | Retains existing trees on site; built-up areas covered with green roofs |
| **Local Materials** | Stone aggregate plaster (grit) | Red sandstone | Lakhori bricks | Compressed-earth blocks | Bamboo and Delhi quartzite stone |
| **Technological Sustainability** | Aggressive use of cantilevered balconies and staircases | Prefabricated waffle slabs create a clear span of 10m | Large sunscreen pergolas have sailcloth panels suspended from a space frame | Hybrid air-conditioning system developed to reduce energy consumption | Use of translucent tensile membrane canopies |
| **Cultural Appropriateness** | Housing units overlook each other's semi private areas to enable communal interaction | Courtyards and roof terraces facilitate communal interaction | Pleasant civic spaces encourage interaction between professionals | Privacy afforded to workspaces; domestic scale in office building | Intimate spaces built to human scale and organised around courtyards and paved allies |

A wide variety of sources have influenced architects in their interpretation of lessons from the past. From organisation of building blocks based on layouts of typical Indian villages or *bazaars* to interlinking of courtyards based on sixteenth-century Mughal imperial palaces, ideas from traditional architecture have helped architects in dealing with contemporary conditions. Each architect has chosen the past on which he wishes to draw upon as per the exigencies of each design problem.

Even though advance mechanised devices for climate-control have been available, regionalist architects have largely relied upon passive means of climate-control. Thick walls, deep balconies, sunscreens, pergolas and carefully-designed fenestrations have been frequently used in regionalist designs in Delhi for climatic mediation.

As most of the building sites located in the Delhi region are flat and even, the topography of sites has not posed a significant challenge to regionalist architects. Wherever possible, regionalist architects have tried to conserve the existing trees on site. As the building codes have not afforded enough room for preserving the existing vegetation on site, the architects have reclaimed the green cover by providing features such as garden courtyards, green roofs and vertical gardens.

As a variety of indigenous materials are readily available in the Delhi region, regionalist architects have been able to choose from an array of building materials to fulfil the economic, aesthetic and ecological concerns specific to a building programme. Labour-intensive construction methods using local materials have also improved employment opportunities at the regional level.

Critical regionalism extends the discourse on sustainable architecture beyond the prevailing technocratic approach by adequately addressing qualitative issues of regional and cultural appropriateness. While technological advancements may accelerate progress, critical reflection is required to understand the capabilities and culpabilities of new technology. Regionalist designs in Delhi reveal how imported technologies have been effectively localised to suit the needs of the community. Another important consideration for sustainable architecture is that a building should 'remain relevant and functional for as long as possible' [41]. Without cultural awareness, energy efficient buildings are likely to 'suffer premature obsolescence and invite major modification or outright demolition and replacement, undermining ambitions for sustainability' [42]. As due care has been taken in the regionalist works in Delhi to address the cultural needs and values of users, they are destined to last longer and be more sustainable in the long run.

In conclusion, it can be said that critical regionalism is not a set of aesthetic preferences but a philosophical framework permissive of a wide variety of viewpoints. Even within a region having uniform climatic conditions, culture, historical context and availability of materials, critical regionalism is capable of generating diverse forms of architecture.

**Author Contributions:** Conceptualisation, Data Curation, Formal Analysis, Investigation, Methodology, Visualisation and Draft Writing by S.B.; Project administration, Supervision and Writing Review by G.R.

**Funding:** This research was funded by the Government of India's Ministry of Human Resource Development through the IIT-Roorkee Doctoral Scholarship to the main author, grant number 14902005.

**Acknowledgments:** Special thanks to the two anonymous reviewers of this manuscript for their helpful comments and suggestions.

**Conflicts of Interest:** The authors declare no conflict of interest.

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
