# Peer review of "A Study of Regional Assertions in the Architecture of Delhi from the 1970s to the present"

_buildings, doi:10.3390/buildings9050108_

Round 1

Reviewer 1 Report

The study entitled Study of Regional Assertions in the Architecture of Delhi from the 1970s to the present, presents five key architectural works realised in Delhi in the past four decades that incorporate the ideas of critical regionalism in their designs.

The study is well structured and well conducted. The authors have organized their material very well, and they elaborate in detail to their audience the definition of the critical regionaalim, as well as how this is incorporated into the 5 projects presented in Delhi.

The use of the English language is appropriate, the referencing of the authors is sufficient, and all statements are well justified and elaborated.

What the study still misses is a more detailed discussion on the current challenges of the critical regionalism, to satisfy the recent advancements in the building sector concerning the sustainability aspects of the construction industry.

Author Response

Point 1: The study entitled Study of Regional Assertions in the Architecture of Delhi from the 1970s to the present, presents five key architectural works realised in Delhi in the past four decades that incorporate the ideas of critical regionalism in their designs.

The study is well structured and well conducted. The authors have organized their material very well, and they elaborate in detail to their audience the definition of the critical regionalism, as well as how this is incorporated into the 5 projects presented in Delhi.

The use of the English language is appropriate, the referencing of the authors is sufficient, and all statements are well justified and elaborated.

What the study still misses is a more detailed discussion on the current challenges of the critical regionalism, to satisfy the recent advancements in the building sector concerning the sustainability aspects of the construction industry.

Response 1: We appreciate the positive feedback from the reviewer. With regards to the sustainability aspect of critical regionalism, we have added a paragraph in the ‘4. Discussion’ section of the revised manuscript discussing how critical regionalism and sustainable architecture are complementary to each other, each essential for the other’s existence. Also discussed is how critical regionalism helps manage risks associated with new technologies.

Reviewer 2 Report

The text is well organized and the research is well structured. The examples are selected to reflect the conclusive points and the research aim of the paper. A comment on the affect of the origin of the Architect might be added on the success of the designs in reflecting the local culture and diversifying the end product. However I am also aware that this is beyond the aims of this study. 

There are a few points to address in order to make the text a little more consistent with the references one is on the first page under the section "2. Critical Regionalism: Definition" the references to Frampton (1985) and Tzonis and Lefaivre are missing in the reference (they are mentioned, but with different dates). A similar referencing problem is existing for the figure captions: the name of the author, date and page numbers are given under the figures but not the reference numbers. This is obviously not a mistake but in order to make the text more consisting and flowing it should use the same format.   

Author Response

Point 1: The text is well organized and the research is well structured. The examples are selected to reflect the conclusive points and the research aim of the paper. A comment on the effect of the origin of the Architect might be added on the success of the designs in reflecting the local culture and diversifying the end product. However, I am also aware that this is beyond the aims of this study.

Response 1: We appreciate the positive feedback from the reviewer. With regards to the origin of the architects, we have added a comment on the effect of the origin of each of the five architects in the revised manuscript. The five comments can be found at lines 94, 200, 300, 415 and 552 of the revised manuscript.

Point 2: There are a few points to address in order to make the text a little more consistent with the references one is on the first page under the section "2. Critical Regionalism: Definition" the references to Frampton (1985) and Tzonis and Lefaivre are missing in the reference (they are mentioned, but with different dates).

Response 2: As suggested by the reviewer, we have added two references to the essays by Alexander Tzonis, Liane Lefaivre and Kenneth Frampton. The two added references can be found at line 43 of the revised manuscript.

Point 3: A similar referencing problem is existing for the figure captions: the name of the author, date and page numbers are given under the figures but not the reference numbers. This is obviously not a mistake but in order to make the text more consisting and flowing it should use the same format.

Response 3: As suggested by the reviewer, we have now added reference numbers to figure captions making them consistent with the referencing format of the rest of the manuscript. The added reference numbers can be found in the captions for figures 3a, 4, 6, 12, 14, 15a, 15b and 16a.